# Understanding Weight Normalized Deep Neural Networks with Rectified Linear Units

**Yixi Xu**
Department of Statistics
Purdue University
West Lafayette, IN 47907
xu573@purdue.edu

**Xiao Wang**
Department of Statistics
Purdue University
West Lafayette, IN 47907
wangxiao@purdue.edu

## Abstract

This paper presents a general framework for norm-based capacity control for $L_{p,q}$ weight normalized deep neural networks. We establish the upper bound on the Rademacher complexities of this family. With an $L_{p,q}$ normalization where $q \leq p^*$ and $1/p + 1/p^* = 1$, we discuss properties of a width-independent capacity control, which only depends on the depth by a square root term. We further analyze the approximation properties of $L_{p,q}$ weight normalized deep neural networks. In particular, for an $L_{1,\infty}$ weight normalized network, the approximation error can be controlled by the $L_1$ norm of the output layer, and the corresponding generalization error only depends on the architecture by the square root of the depth.

## 1 Introduction

During the past decade, deep neural networks (DNNs) have demonstrated an amazing performance in solving many complex artificial intelligence tasks such as object recognition and identification, text understanding and translation, question answering, and more [11]. The capacity of *unregularized* fully connected DNNs, as a function of the network size and depth, is fairly well understood [1, 4, 23]. By bounding the $L_2$ norm of the incoming weights of each unit, [22] is able to accelerate the convergence of stochastic gradient descent optimization across applications in supervised image recognition, generative modeling, and deep reinforcement learning. However, theoretical investigations on such networks are less explored in the literature, and a few exceptions are [4, 5, 10, 18, 19, 25]. There is a central question waiting for an answer: Can we bound the capacity of fully connected DNNs with bias neurons by weight normalization alone, which has the least dependence on the architecture?

In this paper, we focus on networks with rectified linear units (ReLU) and study a more general weight normalized deep neural network (WN-DNN), which includes all layer-wise $L_{p,q}$ weight normalizations. In addition, these networks have a bias neuron per hidden layer, while prior studies [4, 5, 10, 18, 19, 25] either exclude the bias neuron, or only include the bias neuron in the input layer, which differs from the practical application. We establish the upper bound on the Rademacher complexities of this family and study the theoretical properties of WN-DNNs in terms of the approximation error.

We first examine how the $L_{p,q}$ WN-DNN architecture influences their generalization properties. Specifically, for $L_{p,q}$ normalization where $q \leq p^*$ and $1/p + 1/p^* = 1$, we obtain a complexity bound that is independent of width and only has a square root dependence on the depth. To the best of our knowledge, this is the first theoretical result for the fully connected DNNs including a bias neuron for each hidden layer in terms of generalization. We will demonstrate later that it is nontrivial to extend the existing results to the DNNs with bias neurons. Even excluding the bias neurons, existing generalization bounds for DNNs depend on either width or depth logarithmically [5], polynomially[10, 18], or even exponentially [19, 25]. Even for [5], the logarithmic dependency

is not always guaranteed, as the margin bound is

$$O\left(\log(\max\boldsymbol{d})/\sqrt{n}\prod_{i=1}^{k}\|\mathbf{W}_i\|_\sigma\left(\sum_{i=1}^{k}\left\|\mathbf{W}_i^T-\mathbf{M}_i^T\right\|_{2,1}^{2/3}/\|\mathbf{W}_i\|_\sigma^{2/3}\right)^{3/2}\right),$$

where $\|\cdot\|_\sigma$ is the spectral norm, and $\mathbf{M}_i$ is a collection of predetermined reference matrix. The bound will worsen, when the $\mathbf{W}_i$ moves farther from $\mathbf{M}_i$. For example, if

$$\left\|\mathbf{W}_i^T-\mathbf{M}_i^T\right\|_{2,1}/\|\mathbf{W}_i\|_\sigma\geq A_0$$

for some constant $A_0$, then the above bound will rely on the network size by $O\left(\log(\max\boldsymbol{d})k^{3/2}\right)$.

We also examine the approximation error of WN-DNNs. It is shown that the $L_{1,\infty}$ WN-DNN is able to approximate any Lipschitz continuous function arbitrarily well by increasing the norm of its output layer and growing its size. Early work on neural network approximation theory includes the universal approximation theorem [8, 13, 20], indicating that a fully connected network with a single hidden layer can approximate any continuous functions. More recent work expands the result of shallow networks to deep networks with an increased interest in the expressive power of deep networks especially for some families of "hard" functions [2, 9, 16, 21, 26, 27]. For instance, [26] shows that for any positive integer $l$, there exist neural networks with $\Theta(l^3)$ layers and $\Theta(1)$ nodes per layer, which can not be approximated by networks with $\Theta(l)$ layers unless they possess $\Omega(2^l)$ nodes. These results on the other hand request for an artificial neural network of which the generalization bounds grow slowly with depth and even avoid explicit dependence on depth.

The contributions of this paper are summarized as follows.

1. We extend the $L_{2,\infty}$ weight normalization [22] to the more general $L_{p,q}$ WN-DNNs and relate these classes to those represented by unregularized DNNs.

2. We include a bias node not only in the input layer but also in every hidden layer. As discussed in Claim 1, it is nontrivial to extend prior research to study this case.

3. We study the Rademacher complexities of WN-DNNs. Especially, with any $L_{p,q}$ normalization satisfying that $q \leq p^*$, we have a capacity control that is independent of the width and depends on the depth by $O(\sqrt{k})$.

4. We analyze the approximation property of $L_{p,q}$ WN-DNNs and further show the theoretical advantage of $L_{1,\infty}$ WN-DNNs.

The paper is organized as follows. In Section 2, we define the $L_{p,q}$ WN-DNNs and analyze the corresponding function class. Section 3 gives the Rademacher complexities. In Section 4, we provide the error bounds for the approximation error of Lipschitz continuous functions.

## 2   Preliminaries

In this section, we define the WN-DNNs, of which the weights and biases for all layers are scaled by some norm up to a normalization constant $c$. Furthermore, we demonstrate how it surpasses unregularized DNNs theoretically.

A neural network on $\mathbb{R}^{d_0}\to\mathbb{R}^{d_{k+1}}$ with $k$ hidden layers is defined by a set of $k+1$ affine transformations $T_1:\mathbb{R}^{d_0}\to\mathbb{R}^{d_1}, T_2:\mathbb{R}^{d_1}\to\mathbb{R}^{d_2},\cdots,T_{k+1}:\mathbb{R}^{d_k}\to\mathbb{R}^{d_{k+1}}$ and the ReLU activation $\sigma(u)=(u)_+=uI\{u>0\}$. The affine transformations are parameterized by $T_i(\mathbf{u})=\boldsymbol{W}_i^T\mathbf{u}+\mathbf{B}_i$, where $\boldsymbol{W}_i\in\mathbb{R}^{d_{i-1}\times d_i},\mathbf{B}_i\in\mathbb{R}^{d_i}$ for $i=1,\cdots,k+1$. The function represented by this neural network is

$$f(x)=T_{k+1}\circ\sigma\circ T_k\circ\cdots\circ\sigma\circ T_1\circ\boldsymbol{x}$$

Before introducing $L_{p,q}$ WN-DNNs, we build an augmented layer for each hidden layer by appending the bias neuron 1 to the original layer, then combine the weight matrix and the bias vector as a new matrix.

Define $f_0^*(\boldsymbol{x})=(1,\boldsymbol{x}^T)^T$. Then the first hidden layer

$$f_1(\boldsymbol{x})=T_1\circ\boldsymbol{x}\triangleq\tilde{\mathbf{V}}_1^Tf_0^*(\boldsymbol{x}),$$

where $\tilde{\mathbf{V}}_1 = (\mathbf{B}_1, \boldsymbol{W}_1^T)^T \in \mathbb{R}^{(d_0+1) \times d_1}$. Define the augmented first hidden layer as

$$f_1^*(\boldsymbol{x}) = (1, (f_1(\boldsymbol{x}))^T)^T \in \mathbb{R}^{d_1+1}.$$

Then $f_1^*(\boldsymbol{x}) \triangleq \mathbf{V}_1^T f_0^*(\boldsymbol{x})$, where $\mathbf{V}_1 = (\boldsymbol{e}_{10}, \tilde{\mathbf{V}}_1) \in \mathbb{R}^{(d_0+1) \times (d_1+1)}$ and $\boldsymbol{e}_{10} = (1, 0, \cdots, 0)^T \in \mathbb{R}^{d_0+1}$. Sequentially for $i = 2, \cdots, k$, define the $i$th hidden layer as

$$f_i(\boldsymbol{x}) = T_i \circ \sigma \circ f_{i-1}(\boldsymbol{x}) \triangleq \langle \tilde{\mathbf{V}}_i, \sigma \circ f_{i-1}^*(\boldsymbol{x}) \rangle, \tag{1}$$

where $\tilde{\mathbf{V}}_i = (\mathbf{B}_i, \boldsymbol{W}_i^T)^T \in \mathbb{R}^{(d_{i-1}+1) \times d_i}$. Note that $\sigma(1) = 1$, thus $(1, \sigma \circ f_{i-1}(\boldsymbol{x})) = \sigma \circ f_{i-1}^*(\boldsymbol{x})$. The augmented $i$th hidden layer is

$$f_i^*(\boldsymbol{x}) = (1, (f_i(\boldsymbol{x}))^T)^T \in \mathbb{R}^{d_i+1}, \tag{2}$$

and $f_i^*(\boldsymbol{x}) \triangleq \langle \mathbf{V}_i, \sigma \circ f_{i-1}^*(\boldsymbol{x}) \rangle$, where

$$\mathbf{V}_i = (\boldsymbol{e}_{1i}, \tilde{\mathbf{V}}_i) \in \mathbb{R}^{(d_{i-1}+1) \times (d_i+1)}, \tag{3}$$

and $\boldsymbol{e}_{1i} = (1, 0, \cdots, 0)^T \in \mathbb{R}^{d_{i-1}+1}$. The output layer is

$$f(\boldsymbol{x}) = T_{k+1} \circ \sigma \circ f_k^*(\boldsymbol{x}) \triangleq \langle \tilde{\mathbf{V}}_{k+1}, \sigma \circ f_k^*(\boldsymbol{x}) \rangle, \tag{4}$$

where $\tilde{\mathbf{V}}_{k+1} = (\mathbf{B}_{k+1}, \boldsymbol{W}_{k+1}^T)^T \in \mathbb{R}^{(d_k+1) \times d_{k+1}}$.

**The $Lp, q$ Norm.** The $Lp, q$ norm of a $s_1 \times s_2$ matrix $A$ is defined as

$$\|A\|_{p,q} = \left( \sum_{j=1}^{s_2} \left( \sum_{i=1}^{s_1} |a_{ij}|^p \right)^{q/p} \right)^{1/q},$$

where $1 \leq p < \infty$ and $1 \leq q \leq \infty$. When $q = \infty$, $\|A\|_{p,\infty} = \sup_j \left( \sum_{i=1}^{s_1} |a_{ij}|^p \right)^{1/p}$. When $p = q = 2$, the $L_{p,q}$ is the Frobenius norm.

We motivate our introduction of WN-DNNs with a negative result when directly applying existing studies on fully connected DNNs with bias neurons.

**A Motivating Example.** As shown in Figure 1a, define $f = T_2 \circ \sigma \circ T_1 : \mathbb{R} \to \mathbb{R}$, where $T_1(x) = (-x+1, -x-1) \triangleq \tilde{\mathbf{V}}_1^T (1, x)^T$ and $T_2(\mathbf{u}) = 1 - u_1 - u_2 \triangleq \tilde{\mathbf{V}}_2^T (1, u_1, u_2)^T$. Consider $f' = 100 T_2 \circ \sigma \circ \frac{1}{100} T_1$, as shown in Figure 1b . Then

$$f'(x) = 100 - \sigma(-x+1) - \sigma(-x-1) = 99 + f(x)$$

Note that the product of the norms of all layers for $f'$ remains the same as that for $f$:

$$\|100 T_2\|_* * \left\| \frac{T_1}{100} \right\|_* = \|T_2\|_* * \|T_1\|_* ,$$

where the norm of the affine transformation $\|T_i\|_*$ is defined as the norm of its corresponding linear transformation matrix $\left\| \tilde{\mathbf{V}}_i \right\|_*$ for $i = 1, 2$. Using a similar trick, we could replace the 100 in this example with any positive number. This on the other hand suggests an unbounded output even when the product of the norms of all layers is small.

Furthermore, a negative result will be presented in terms of Rademacher complexity in the following claim.

**Claim 1.** *Define $\mathcal{N}_{\gamma_* \leq \gamma}^{k, \boldsymbol{d}}$ as a function class that contains all functions representable by some neural network of depth $k+1$ and widths $\boldsymbol{d}$: $f = T_{k+1} \circ \sigma \circ T_k \circ \cdots \circ \sigma \circ T_1 \circ \boldsymbol{x}$, where $\boldsymbol{d} = (m_1, d_1, \cdots, d_k, 1)$, $\|\cdot\|_*$ is an arbitrary norm, and $T_i(\mathbf{u}) : \mathbb{R}^{d_{i-1}} \to \mathbb{R}^{d_i} = \tilde{\mathbf{V}}_i^T (1, \mathbf{u}^T)^T$, for $i = 1, \cdots, k+1$, such that*

$$\gamma_* = \prod_{i=1}^{k+1} \left\| \tilde{\mathbf{V}}_i \right\|_* \leq \gamma.$$

*Then for a fixed n and any sample $S = \{\boldsymbol{x}_1, \cdots, \boldsymbol{x}_n\} \subseteq \mathbb{R}^{m_1}$,*

$$\widehat{\mathfrak{R}}_S(\mathcal{N}_{\gamma_* \leq \gamma}^{k, \boldsymbol{d}}) = \infty.$$

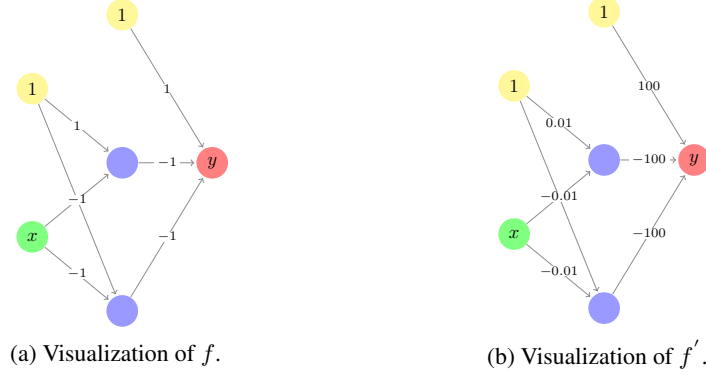

(a) Visualization of $f$.　　　　　(b) Visualization of $f'$.

Figure 1: The motivating example.

Claim 1 shows the failure of current norm-based constraints on fully connected neural networks with the bias neuron in each hidden layer. Prior studies [4, 5, 10, 18, 19, 25] included the bias neuron only in the input layer and considered layered networks parameterized by a sequence of weight matrices only, that is $\mathbf{B}_i = \mathbf{0}$ for all $i = 1, \cdots, k+1$. While fixing the architecture of neural networks, these works imply that $\prod_{i=1}^{k+1} \|\mathbf{W}_i\|_*$ is sufficient to control the Rademacher complexity of the function class represented by these DNNs, where $\|\cdot\|_*$ is the spectral norm in [5, 18], the $L_{1,\infty}$ norm in [4, 25], the $L_{1,\infty}/L_{2,2}$ norm in [10], and the $L_{p,q}$ norm in [19] for any $p \in [1, \infty), q \in [1, \infty]$. However, this kind of control fails once the bias neuron is added to each hidden layer, demonstrating the necessity to use WN-DNNs instead.

**The $L_{p,q}$ WN-DNNs.**　An $L_{p,q}$ WN-DNN by a normalization constant $c \geq 1$ with $k$ hidden layers is defined by a set of $k+1$ affine transformations $T_1 : \mathbb{R}^{d_0} \to \mathbb{R}^{d_1}, T_2 : \mathbb{R}^{d_1} \to \mathbb{R}^{d_2}, \cdots, T_{k+1} : \mathbb{R}^{d_k} \to \mathbb{R}^{d_{k+1}}$ and the ReLU activation, where $T_i(\mathbf{u}) = \tilde{\mathbf{V}}_i^T (1, \mathbf{u}^T)^T$, $\tilde{\mathbf{V}}_i \in \mathbb{R}^{(d_{i-1}+1) \times d_i}$ and $\|T_i\|_{p,q} \triangleq \left\| \tilde{\mathbf{V}}_i \right\|_{p,q}$, for $i = 1, \cdots, k+1$. In addition, $\|T_i\|_{p,q} \equiv c$ for $i = 1, \cdots, k$.

Define $\mathcal{N}_{p,q,c,c_o}^{k,\boldsymbol{d}}$ as the collection of all functions that could be represented by an $L_{p,q}$ WN-DNN with the normalization constant $c$ satisfying:

(a) The number of neurons in the $i$th hidden layer is $d_i$ for $i = 1, 2, \cdots, k$. The dimension of input is $d_0$, and output $d_{k+1}$;

(b) It has $k$ hidden layers;

(c) $\|T_i\|_{p,q} \equiv c$ for $i = 1, \cdots, k$;

(d) $\|T_{k+1}\|_{p,q} \leq c_o$.

The following theorem provides some useful observations regarding $\mathcal{N}_{p,q,c,c_o}^{k,\boldsymbol{d}}$.

**Theorem 1.** *Let $c, c_o, c_1, c_2, c_o^1, c_o^2 > 0$, $p \in [1, \infty)$, $q \in [1, \infty]$, $k, k_1, k_2 \in \mathbb{N}$, $\boldsymbol{d} = (d_0, d_1 \cdots, d_{k+1}) \in \mathbb{N}_+^{k+2}$, $\boldsymbol{d}^1 = (d_0^1, d_1^1 \cdots, d_{k_1+1}^1) \in \mathbb{N}_+^{k_1+2}$, and $\boldsymbol{d}^2 = (d_0^2, d_1^2 \cdots, d_{k_2+1}^2) \in \mathbb{N}_+^{k_2+2}$.*

*(a) A function $f : \mathbb{R}^{d_0} \to \mathbb{R}^{d_{k+1}} = T_{k+1} \circ \sigma \circ T_k \circ \cdots \circ \sigma \circ T_1 \circ \boldsymbol{x}$, where $T_i(\mathbf{u}) = \boldsymbol{W}_i^T \mathbf{u} + \mathbf{B}_i : \mathbb{R}^{d_{i-1}} \to \mathbb{R}^{d_i}$. Then $f \in \mathcal{N}_{p,q,c,c_o}^{k,\boldsymbol{d}}$, as long as $\|T_i\|_{p,q} \leq c$ for $i = 1, \cdots, k$ and $\|T_{k+1}\|_{p,q} \leq c_o$.*

*(b) $\mathcal{N}_{p,q,c_1,c_o}^{k,\boldsymbol{d}} \subseteq \mathcal{N}_{p,q,c_2,c_o}^{k,\boldsymbol{d}}$ if $c_1 \leq c_2$. $\mathcal{N}_{p,q,c,c_o^1}^{k,\boldsymbol{d}} \subseteq \mathcal{N}_{p,q,c,c_o^2}^{k,\boldsymbol{d}}$ if $c_o^1 \leq c_o^2$. If $g \in \mathcal{N}_{p,q,c,1}^{k,\boldsymbol{d}}$, then $c_o g \in \mathcal{N}_{p,q,c,c_o}^{k,\boldsymbol{d}}$.*

*(c) $\mathcal{N}_{p_1,q,c,c_o}^{k,\boldsymbol{d}} \subseteq \mathcal{N}_{p_2,q,c,c_o}^{k,\boldsymbol{d}}$ if $1 \leq p_1 \leq p_2 < \infty$. $\mathcal{N}_{p,q_1,c,c_o}^{k,\boldsymbol{d}} \subseteq \mathcal{N}_{p,q_2,c,c_o}^{k,\boldsymbol{d}}$ if $1 \leq q_1 \leq q_2 \leq \infty$.*

$\mathcal{N}^{k,\boldsymbol{d}}_{p,\infty,c,c_o} \subseteq \mathcal{N}^{k,\boldsymbol{d}}_{p,q,\tilde{c},\tilde{c}_o}$, where $\tilde{c} = c \max^{\frac{1}{q}}\{d_1, d_2 \cdots, d_k\}$ and $\tilde{c}_o = d^{\frac{1}{q}}_{k+1}c_o$. Especially, when $d_{k+1} = 1$, $\tilde{c}_o = c_o$.

(d) $\mathcal{N}^{k_1,\boldsymbol{d}^1}_{p,q,c,c_o} \subseteq \mathcal{N}^{k_2,\boldsymbol{d}^2}_{p,q,c,c_o}$ if $c \geq 1$, $k_1 \leq k_2$, $d^2_0 = d^1_0$, $d^2_i \geq d^1_i$ for $i = 1, \cdots, k_1$, $d^2_i \geq d^1_{k_1+1}$ for $i > k_1$, and $d^2_{k_2+1} = d^1_{k_1+1} = 1$.

In particular, Part (a) connects normalized neural networks to unregularized DNNs. Part (b) shows the increased expressive power of neural networks by increasing the normalization constant or the output layer norm constraint. Part (c) discusses the influence of the choice of $L_{p,q}$ normalization on its representation capacity. Part (d) describes the gain in representation power by either widening or deepening the neural networks.

# 3   Estimating the Rademacher Complexities of $\mathcal{N}^{k,\boldsymbol{d}}_{p,q,c,c_o}$

In this section, we bound the Rademacher complexities of $\mathcal{N}^{k,\boldsymbol{d}}_{p,q,c,c_o}$, where $d_0 = m_1$ and $d_{k+1} = 1$. Without loss of generality, assume the input space $\mathcal{X} = [-1,1]^{m_1}$ in the following sections. Further define $p^*$ by $1/p + 1/p^* = 1$.

**Proposition 1.** *Fix* $q \geq 1, k \geq 0, c, c_o > 0, d_i \in \mathbb{N}_+$ *for* $i = 1, \cdots, k$, *then for any set* $S = \{\boldsymbol{x}_1, \cdots, \boldsymbol{x}_n\} \subseteq \mathcal{X}$, *we have*

$$\widehat{\mathfrak{R}}_S(\mathcal{N}^{k,\boldsymbol{d}}_{1,q,c,c_o}) \leq \frac{c_o}{\sqrt{n}} \min\left(2\max(1, c^k)\sqrt{k + 2 + \log(m_1 + 1)}, \right.$$
$$\left. \sqrt{k\log 16}\sum_{i=0}^{k} c^i + c^k(\sqrt{2\log(2m_1)} + \sqrt{k\log 16})\right).$$

*Proof sketch.* As $\sigma(1) = 1$, we could treat the bias neuron in the $i$th hidden layer as a hidden neuron computed from the $(i-1)$th hidden layer by

$$\sigma(\boldsymbol{e}^T_{1i}f^*_{i-1}(\boldsymbol{x})) = 1,$$

where $\boldsymbol{e}_{1i} = (1, 0, \cdots, 0)^T \in \mathbb{R}^{d_{i-1}+1}$, and $f^*_{i-1}$ is the augmented $(i-1)$th hidden layer as defined in Equation (2). Therefore, the new affine transformation could be parameterized by $\mathbf{V}_i$ defined in Equation (3), such that $\|\mathbf{V}_i\|_{1,\infty} = \max(1, c)$. Then the result is the minimum of the bound of [10, Theorem 2] on DNNs without bias neurons and that of Proposition 2 when $p = 1$.   $\square$

**Proposition 2.** *Fix* $p, q \geq 1, k \geq 0, c, c_o > 0, d_i \in \mathbb{N}_+$ *for* $i = 1, \cdots, k$, *then for any set* $S = \{\boldsymbol{x}_1, \cdots, \boldsymbol{x}_n\} \subseteq \mathcal{X}$, *we have*

(a) *for* $p \in (1, 2]$,

$$\widehat{\mathfrak{R}}_S(\mathcal{N}^{k,\boldsymbol{d}}_{p,q,c,c_o}) \leq c_o\sqrt{\frac{(k+1)\log 16}{n}} \left(\sum_{i=1}^{k+1} c^{k-i+1}\prod_{l=i}^{k} d_l^{[\frac{1}{p^*}-\frac{1}{q}]_+}\right) +$$
$$\frac{c_o c^k}{\sqrt{n}}\prod_{i=1}^{k} d_i^{[\frac{1}{p^*}-\frac{1}{q}]_+} m_1^{\frac{1}{p^*}}\left[\min\left((\sqrt{p^*-1}, \sqrt{2\log(2m_1)}\right) + \sqrt{(k+1)\log 16}\right],$$
$$(5)$$

(b) *for* $p \in 1 \cup (2, \infty)$,

$$\widehat{\mathfrak{R}}_S(\mathcal{N}^{k,\boldsymbol{d}}_{p,q,c,c_o}) \leq c_o\sqrt{\frac{(k+1)\log 16}{n}} \left(\sum_{i=1}^{k+1} c^{k-i+1}\prod_{l=i}^{k} d_l^{[\frac{1}{p^*}-\frac{1}{q}]_+}\right) +$$
$$\frac{1}{\sqrt{n}}c_o c^k\prod_{i=1}^{k} d_i^{[\frac{1}{p^*}-\frac{1}{q}]_+} m_1^{\frac{1}{p^*}}\left(\sqrt{2\log(2m_1)} + \sqrt{(k+1)\log 16}\right).$$
$$(6)$$

*Proof sketch.* The proof consists of two steps. In the first step, following the notations in Section 2, we define a series of random variables

$$Z_j = \sup_{f \in \mathcal{N}_{p,q,c,c_o}^{k,\boldsymbol{d}}} \left\| \sum_{i=1}^{n} \epsilon_i \sigma \circ f_j(\boldsymbol{x}_i) \right\|_{p^*},$$

where $\{\epsilon_1, \cdots, \epsilon_n\}$ are $n$ i.i.d Rademacher random variables, and $f_j$ is the $j$th hidden layer of the neural network $f$. We prove by induction that for any $t \in \mathbb{R}$,

$$\mathbb{E}_\epsilon \exp(tZ_j) \le 4^j \exp\left( \frac{t^2 n s_j^2}{2} + tc^j \prod_{i=1}^{j} d_i^{[\frac{1}{p^*} - \frac{1}{q}]_+} A_{m_1,S}^p \right),$$

where

$$s_j = \sum_{i=2}^{j} c^{j-i+1} \prod_{l=i}^{j} d_l^{[\frac{1}{p^*} - \frac{1}{q}]_+} + (m_1^{1/p^*} + 1)c^j \prod_{l=1}^{j} d_l^{[\frac{1}{p^*} - \frac{1}{q}]_+},$$

and $A_{m_1,S}^p$ is some constant only depends on the sample. In addition, we relies on Hölder's inequality with an optimal parameter to separate the bias neuron. Step 2 is motivated by the idea of [10]. By Jensen's inequality

$$n\widehat{\mathfrak{R}}_S(\mathcal{N}_{p,q,c,c_o}^{k,\boldsymbol{d}}) \le \frac{1}{\lambda} \log \mathbb{E}_\epsilon \exp\left( \lambda \sup_{f \in \mathcal{N}_{p,q,c,c_o}^{k,\boldsymbol{d}}} \left( \sum_{i=1}^{n} \epsilon_i f(\boldsymbol{x}_i) \right) \right).$$

Finally we get the desired result by choosing the optimal $\lambda$. □

When $\boldsymbol{d} = d\mathbf{1}$, the upper bound of Rademacher complexity depends on the width by $O(d^{k[\frac{1}{p^*} - \frac{1}{q}]_+})$, which is similar to the case without bias neurons [19]. Furthermore, the dependence on widths disappears as long as $q \in [1, p^*]$. In order to investigate the tightness of the bound given in Proposition 2, we consider the binary classification as a specific case, indicating that when $\frac{1}{p} + \frac{1}{q} < 1$, the dependence on width is unavoidable.

**Proposition 3.** *[19, Theorem 3] For any $p, q \ge 1$, $\boldsymbol{d} = d\mathbf{1}$ and any $k \ge 2$, $n$ $\{-1, +1\}$ points could be shattered with unit margin by $\mathcal{N}_{p,q,c,c_o}^{k,\boldsymbol{d}}$, with*

$$c^k c_o \le (\log_2 n)^{\frac{1}{p}} n^{(\frac{1}{p} + \frac{1}{q})} d^{-(k-2)[\frac{1}{p^*} - \frac{1}{q}]_+}.$$

**Issues on Bias Neurons.** $L_{p,q}$ norm-constrained fully connected DNNs with no bias neuron were investigated in prior studies [4, 10, 19, 25]. First of all, the generalization bounds given by [4, 19, 25] have explicit exponential dependence on the depth, thus it is not meaningful to compare these results with ours. Secondly, [10] provides the up-to-date Rademacher complexity bounds of both $L_{1,\infty}$ and $L_{2,2}$ norm-constrained fully connected DNNs without bias neurons. However, it is not straightforward to extend their results to fully connected DNNs with a bias neuron in each hidden layer. For example, consider the $L_{2,2}$ WN-DNNs with $c = 1$. If we simply treat each bias neuron as a hidden neuron, as in the proof for Proposition 1, the complexity bounds [10] grows exponentially with respect to the depth by $O(\sqrt{k}2^{\frac{k}{2}})$, while our Proposition 2 gives a much tighter bound $O(k^{\frac{3}{2}})$.

**Comparison with [10] on the Rademacher compexity bounds of $L_{1,\infty}$ and $L_{2,2}$ WN-DNNs.** [10] is the most recent work on the Rademacher complexities of the $L_{1,\infty}$ and $L_{2,2}$ norm-constrained fully connected DNNs without bias neurons. Consider a specific case when $\log(m_1)$ is small and $c_o = 1$ to shed light on the possible influence of the bias neurons on the generalization properties.

As summarized in Table 1, these comparisons suggest that the inclusion of a bias neuron in each hidden layer might lead to extra dependence of generalization bounds on the depth especially when $c$ is small. Note that, when $c < 1$, $\sqrt{k}(1 - c^{k+1})/(1 - c) \to \infty$, while $\sqrt{k}c^k \to 0$, as $k \to \infty$. For $L_{2,2}$ WN-DNNs, when $c = 1$, the bounds are $O(\frac{k^{\frac{3}{2}}}{\sqrt{n}})$ if with bias neurons and $O(\frac{\sqrt{k}}{\sqrt{n}})$ without bias neurons. For $L_{2,2}$ WN-DNNs, when $c > 1$, the bounds are $O(\frac{\sqrt{k}(c^{k+1}-1)}{\sqrt{n}})$ if including bias neurons and $O(\frac{\sqrt{k}c^k}{\sqrt{n}})$ if excluding bias neurons. Another interesting observation is that the complexity bounds remain the same no matter whether bias neurons are included or not, when $c > 1$ for $L_{1,\infty}$ WNN-DNNs.

| | With Bias Neurons | Without Bias Neurons [10] |
|---|---|---|
| $c < 1$ | $O(\frac{\sqrt{k}(1-c^{k+1})}{(1-c)\sqrt{n}})$ | $O(\frac{\sqrt{k}c^k}{\sqrt{n}})$ |
| $c = 1, L_{1,\infty}$ | $O(\frac{\sqrt{k}}{\sqrt{n}})$ | $O(\frac{\sqrt{k}}{\sqrt{n}})$ |
| $c = 1, L_{2,2}$ | $O(\frac{k^{3/2}}{\sqrt{n}})$ | $O(\frac{\sqrt{k}}{\sqrt{n}})$ |
| $c > 1, L_{1,\infty}$ | $O(\frac{\sqrt{k}c^k}{\sqrt{n}})$ | $O(\frac{\sqrt{k}c^k}{\sqrt{n}})$ |
| $c > 1, L_{2,2}$ | $O(\frac{\sqrt{k}(c^{k+1}-1)}{\sqrt{n}})$ | $O(\frac{\sqrt{k}c^k}{\sqrt{n}})$ |

Table 1: Rademacher complexity bounds for $L_{1,\infty}/L_{2,2}$ WN-DNNs with/without bias neurons.

## 4  Approximation Properties

In this section, we analyze the approximation properties of $L_{p,q}$ WN-DNNs and show the theoretical advantage of $L_{1,\infty}$ WN-DNN. We first introduce a technical lemma, demonstrating that any wide one-hidden-layer neural network could be exactly represented by a deep but narrow normalized neural network. In addition, Lemma 1 indicates that $\mathcal{N}_{1,\infty,\cdot,c_o}^{1,(m_1,r,1)} \subseteq \mathcal{N}_{p,\infty,1,2c_o}^{k,(m_1,([r/k]+2m_1+3)\mathbf{1}_k,1)}$ for any $r > 1$, $k \in \mathcal{N}$, and $c_o > 0$, where $[x]$ is the smallest integer which is greater than or equal to $x$, and $\mathbf{1}_k = (1,\cdots,1) \in \mathbb{R}^k$.

**Lemma 1.** *Assume that a function*

$$g(\boldsymbol{x}) : \mathbb{R}^{m_1} \to \mathbb{R} = \sum_{i=1}^{r} c_i \sigma(\boldsymbol{w}_i^T \boldsymbol{x} + b_i)$$

*satisfies that* $\sum_{i=1}^{r} |c_i| \leq c_o$ *and* $\left\|(b_i, \boldsymbol{w}_i^T)\right\|_1 = 1$. *Then for any integer* $k \in [1,r]$,

$$g \in \mathcal{N}_{p,q,wid_k^{1/q},2c_o}^{k,\boldsymbol{d}^k},$$

*where* $wid_k = [r/k] + 2m_1 + 3$, $d_0^k = m_1$, $d_i^k = wid_k$ *for* $i = 1,\cdots,k$, *and* $d_{k+1}^k = 1$.

*Proof sketch.* Note that the shallow neural network $g$ could be decomposed as

$$\sum_{i=1}^{r_1} c_i^+ \sigma\left((\boldsymbol{w}_i^+)^T \boldsymbol{x} + b_i^+\right) - \sum_{i=1}^{r_2} c_i^- \sigma\left((\boldsymbol{w}_i^-)^T \boldsymbol{x} + b_i^-\right),$$

where $c_i^+, c_i^- > 0$ and $r_1 + r_2 = r$. We consider a simplified case when $g(\boldsymbol{x}) = \sum_{i=1}^{r_1} c_i^+ \sigma\left((\boldsymbol{w}_i^+)^T \boldsymbol{x} + b_i^+\right)$ to illustrate the main idea of our proof. Without loss of generality, assume that $\left\|(b_i, 2\boldsymbol{w}_i^T)\right\|_1 = 1$. First create a set

$$\mathcal{C} = \{\sigma\left((\boldsymbol{w}_i^+)^T \boldsymbol{x} + b_i^+\right), i = 1,\cdots,r_1\}.$$

In order to build a $k + 1$-layer WN-DNN to represent $g$, we partition $\mathcal{C}$ into $k$ equally sized subsets: $\mathcal{C}_1,\cdots,\mathcal{C}_k$. The key idea is to get all elements of $\mathcal{C}_j$ in the $j$th hidden layer for $j = 1,\cdots,k$, while keeping both $\sigma \circ \boldsymbol{x}$, and $\sigma \circ -\boldsymbol{x}$. In addition, the normalized cumulative sum $S_j$ of $\cup_{i \leq j} \mathcal{C}_i$ is computed in the $j + 1$th hidden layer. More specifically,

$$S_j = \frac{\sum_{i=1}^{jr_1/k} c_i^+ \sigma\left((\boldsymbol{w}_i^+)^T \boldsymbol{x} + b_i^+\right)}{\sum_{i=1}^{jr_1/k} c_i^+}.$$

Note that

$$(\boldsymbol{w}_i^+)^T \boldsymbol{x} + b_i^+ = (\boldsymbol{w}_i^+)^T \sigma \circ \boldsymbol{x} - (\boldsymbol{w}_i^+)^T \sigma \circ (-\boldsymbol{x}) + b_i^+,$$

and

$$S_j = \frac{\sum_{i=1}^{(j-1)r_1/k} c_i^+}{\sum_{i=1}^{jr_1/k} c_i^+} \sigma(S_{j-1}) + \sum_{i=(j-1)r_1/k+1}^{jr_1/k} \frac{c_i^+}{\sum_{i=1}^{jr_1/k} c_i^+} \sigma\left((\boldsymbol{w}_i^+)^T \boldsymbol{x} + b_i^+\right).$$

Thus the $L_{1,\infty}$ norm of the corresponding transformation still $\leq 1$. $\qquad\square$

Based on Lemma 1, we establish that a WN-DNN is able to approximate any Lipschitz-continuous function arbitrarily well by loosing the constraint for the norm of the output layer and either widening or deepening the neural network at the same time. Especially, for $L_{p,\infty}$ WN-DNNs, the approximation error could be purely controlled by the norm of the output layer, while the $L_{p,\infty}$ norm of each hidden layer is fixed to be 1.

**Theorem 2.** $f : \mathcal{X} \to \mathbb{R}$, *satisfying that* $\|f\|_\infty \leq L$, *and* $|f(x) - f(y)| \leq L\|x - y\|_\infty$. *Then for any* $p \in [1,\infty)$, $q \in [1,\infty]$, *and any integer* $k \in [1, C_r(m_1)(\log \frac{c_o}{L})^{-2(m_1+1)/(m_1+4)} \left(\frac{c_o}{L}\right)^{2(m_1+3)/(m_1+4)}]$, *if $c_o$ greater than a constant depending only on $m_1$, there exists a function $h \in \mathcal{N}_{p,q,wid_k^{1/q},2c_o}^{k,\boldsymbol{d}^k}$, where*

$$wid_k = [k^{-1}C_r(m_1)(\log \frac{c_o}{L})^{-\frac{2(m_1+1)}{m_1+4}} \left(\frac{c_o}{L}\right)^{\frac{2(m_1+3)}{m_1+4}}] + 2m_1 + 3,$$

$\boldsymbol{d}^k = (m_1, wid_k, \cdots, wid_k, 1)$, *such that*

$$\sup_{\|\boldsymbol{x}\|_\infty \leq 1} |f(\boldsymbol{x}) - h(\boldsymbol{x})| \leq C(m_1)L(\frac{c_o}{L})^{-\frac{2}{m_1+1}} \log \frac{c_o}{L},$$

*where $C_r(m_1)$ and $C(m_1)$ denotes some constant that depends only on $m_1$.*

Theorem 2 shows that the approximation bounds could be controlled by $c_o$ given a sufficiently deep or wide $L_{p,q}$ WN-DNN. Assume that the loss function is 1-Lipschitz continuous, then the dependence of the corresponding generalization bound on the architecture for each $\mathcal{N}_{p,q,wid_k^{1/q},2c_o}^{k,\boldsymbol{d}^k}$ defined above are summarized as follows:

(a) $p = 1, q = \infty$: $O\left(\sqrt{k}c_o\right)$;

(b) $p = 1, q < \infty$: $O\left(\sqrt{k}c_o wid_k^{\frac{k}{q}}\right)$;

(c) $p > 1, q \in (p^*, \infty]$: $O\left(\sqrt{k}c_o[(1 + wid_k)^{\frac{1}{p^*}}]^k\right)$;

(d) $p > 1, q \in [1, p^*]$: $O\left(\sqrt{k}c_o[(1 + wid_k)^{\frac{1}{q}}]^k\right)$.

## 5 Concluding Remarks

We present a general framework for capacity control on WN-DNNs. In particular, we provide a satisfying answer for the central question: we obtain the generalization bounds for $L_{1,\infty}$ WN-DNNs that grows with depth by a square root term while getting the approximation error controlled. It will be interesting to extend this work to mullticlass classification. However, if handling via Radermacher complexity analysis, the generalization bound will depend on the square root of the number of classes [28]. Besides the extension to convolutional neural networks, we are also working on the design of effective algorithms for $L_{1,\infty}$ WN-DNNs.

**Acknowledgments**

We thank the anonymous reviewers for their careful reading of our manuscript and their insightful comments that have greatly improved the paper.

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
