[Supplementary Material]

**Supplementary Material**

## A  Claim 1

### A.1  Proof for Claim 1

*Proof.* We first show that for any $\gamma_0 > 0$, any norm $\|\cdot\|_*$, and any $C_0 > 0$, there exists a function $f_{\tilde{\mathbf{V}}}$ satisfying $f_{\tilde{\mathbf{V}}} \equiv C_0$ and $\prod_{i=1}^{k+1} \left\|\tilde{\mathbf{V}}_i^T\right\|_* \leq \gamma_0$. First assume that $\|(1,0,\cdots,0)\|_* = a_0$. Note that $a_0 > 0$ by the definition of the norm. To prove this, we could set an arbitrary $\tilde{\mathbf{V}}_1$ satisfying that $\left\|\tilde{\mathbf{V}}_1^T\right\|_* = \frac{\gamma_0}{a_0 C_0}$, the arbitrary $\tilde{\mathbf{V}}_i$s satisfying that $\left\|\tilde{\mathbf{V}}_i\right\|_* = 1$ for $i = 2,\cdots,k$, and the output layer as $T_{k+1}(\mathbf{u}) = C_0$. Then $f_{\tilde{\mathbf{V}}} \equiv C_0$, and

$$\prod_{i=1}^{k+1} \left\|\tilde{\mathbf{V}}_i\right\|_* \leq \frac{\gamma_0}{a_0 C_0} * 1^{k-1} * a_0 C_0 = \gamma_0.$$

Then

$$\widehat{\mathfrak{R}}_S(\mathcal{N}_{\gamma_* \leq \gamma}^{k,\boldsymbol{d}}) = \mathbb{E}_\epsilon \left[ \sup_{f \in \mathcal{F}} \left( \frac{1}{n} \sum_{i=1}^n \epsilon_i f(z_i) \right) \right]$$

$$\geq \mathbb{P}(\sum_{i=1}^n \epsilon_i \neq 0) \mathbb{E}_\epsilon \left[ \sup_{f \in \mathcal{N}_{\gamma_* \leq \gamma}^{k,\boldsymbol{d}}} \left( \frac{1}{n} \sum_{i=1}^n \epsilon_i f(z_i) \right) | \sum_{i=1}^n \epsilon_i \neq 0 \right]$$

$$\geq \frac{1}{2} \mathbb{E}_\epsilon \left[ \sup_{f \in \mathcal{N}_{\gamma_* \leq \gamma}^{k,\boldsymbol{d}}} \left( \frac{1}{n} \sum_{i=1}^n \epsilon_i f(z_i) \right) | \sum_{i=1}^n \epsilon_i \neq 0 \right] \qquad (7a)$$

$$\geq \frac{1}{2} \mathbb{E}_\epsilon \left[ \sup_{C_0 > 0} \left( \frac{1}{n} \sum_{i=1}^n \epsilon_i sgn(\sum_{i=1}^n \epsilon_i) C_0 \right) | \sum_{i=1}^n \epsilon_i \neq 0 \right]$$

$$= \infty,$$

where the step in Equation (7a) follows from $\mathbb{P}(\sum_{i=1}^n \epsilon_i \neq 0) = 1$ when $n$ is an odd number, and $\mathbb{P}(\sum_{i=1}^n \epsilon_i \neq 0) = 1 - \frac{1}{2}\mathbb{P}(\sum_{i=2}^n \epsilon_i = 1) - \frac{1}{2}\mathbb{P}(\sum_{i=2}^n \epsilon_i = -1) \geq \frac{1}{2}$ when n is an even number.  □

## B  Theorem 1

*Proof.* For Part (a), if any $\|T_i\|_{p,q} = 0$, then $f = 0 \in \mathcal{N}_{p,q,c,c_o}^{k,\boldsymbol{d}}$. Otherwise, we will prove by induction on depth $k+1$. It is trivial when $k = 0$.

When $k = 1$, we rescale the first hidden layer by

$$s = c/\|T_1\|_{p,q}.$$

Equivalently, define the new affine transformation $T_1^*$ by

$$\mathbf{B}_1^* = s\mathbf{B}_1, \mathbf{W}_1^* = s\mathbf{W}_1,$$

such that $\|T_1^*\|_{p,q} = c$. For the output layer, we define

$$\mathbf{W}_2^* = \mathbf{W}_2 \|T_1^*\|_{p,q}/c, \mathbf{B}_2^* = \mathbf{B}_2.$$

Then $T_2^*(\mathbf{u}) = (\mathbf{W}_2^*)^T \mathbf{u} + \mathbf{B}_2^*$ satisfies $\|T_2^*(\mathbf{u})\|_{p,q} \leq c_o$, as $s \geq 1$. What's more $f(\boldsymbol{x}) = T_2^* \circ \sigma \circ T_1^* \circ \boldsymbol{x} \in \mathcal{N}_{p,q,c,c_o}^{1,\boldsymbol{d}}$.

Assume the result holds when $k < K$. Then when $k = K$, consider $f(\boldsymbol{x}) = T_{K+1} \circ \sigma \circ T_K \circ \cdots \sigma \circ T_1^* \circ \boldsymbol{x}$. Its $K$th hidden layer

$$f_K(\boldsymbol{x}) \in \mathcal{N}_{p,q,c,c}^{K-1,\boldsymbol{d}_K}$$

by induction assumption, where $\boldsymbol{d}_K = (d_0, d_1 \cdots, d_K)$. In other words, there exists a series of affine transformations $\{T_i^*\}_{i=1,\cdots,K}$, such that

$$f_K(\boldsymbol{x}) = T_K^* \circ \sigma \circ T_{K-1}^* \circ \cdots \circ \sigma \circ T_1^* \circ \boldsymbol{x},$$

$\|T_i^*\| = c$ for $i = 1, \cdots, K-1$, and $\|T_K^*\| \le c$. Thus

$$f(\mathbf{x}) = T_{K+1} \circ \sigma \circ T_K^* \circ \sigma \circ T_{K-1}^* \circ \cdots \circ \sigma \circ T_1^* \circ \boldsymbol{x}.$$

We rescale $T_K^*$ by $s = c/\|T_K^*\|_{p,q}$. Equivalently, define a new affine transformation $T_K^{**}$ by $T_K^{**} = sT_K^*$, such that $\|T_K^{**}\|_{p,q} = c$. For the output layer, we define

$$\mathbf{W}_{K+1}^* = \mathbf{W}_{K+1}/s, \mathbf{B}_{K+1}^* = \mathbf{B}_{K+1}.$$

Then $T_{K+1}^*(\mathbf{u}) = (\mathbf{W}_{K+1}^*)^T\mathbf{u} + \mathbf{B}_{K+1}^*$ satisfies $\left\|T_{K+1}^*(\mathbf{u})\right\|_{p,q} \le c_o$, as $s \ge 1$. Thus $f \in \mathcal{N}_{p,q,c,c_o}^{K,\boldsymbol{d}}$.

For Part (b), it is a direct conclusion from Part (a) that $\mathcal{N}_{p,q,c_1,c_o}^{k,\boldsymbol{d}} \subseteq \mathcal{N}_{p,q,c_2,c_o}^{k,\boldsymbol{d}}$ if $c_1 \le c_2$, and $\mathcal{N}_{p,q,c,c_o^1}^{k,\boldsymbol{d}} \subseteq \mathcal{N}_{p,q,c,c_o^2}^{k,\boldsymbol{d}}$ if $c_o^1 \le c_o^2$. If $g \in \mathcal{N}_{p,q,c,1}^{k,\boldsymbol{d}}$, then by definition, $c_o g \in \mathcal{N}_{p,q,c,c_o}^{k,\boldsymbol{d}}$.

For Part (c), note that $\|\cdot\|_{p_1} \ge \|\cdot\|_{p_2}$ when $p_1 \le p_2$, hence

$$\{\boldsymbol{v} : \|\boldsymbol{v}\|_{p_1} \le C\} \subseteq \{\boldsymbol{v} : \|\boldsymbol{v}\|_{p_2} \le C\}.$$

Then the first line of Part (c) follows from the observation above as well as the conclusion of Part (a). As for the second line, for any $h \in \mathcal{N}_{p,\infty,c,c_o}^{k,\boldsymbol{d}}$, we could write

$$h = T_{k+1} \circ \sigma \circ T_k \circ \cdots \circ \sigma \circ T_1 \circ \boldsymbol{x},$$

where $T_i(\mathbf{u}) : \mathbb{R}^{d_{i-1}} \to \mathbb{R}^{d_i} = \boldsymbol{W}_i^T\mathbf{u} + \mathbf{B}_i$, satisfies that $\|T_i\|_{p,\infty} = c$ for $i = 1, \cdots, k$, and $\|T_{k+1}\|_{p,\infty} \le c_o$. Note that

$$\|T_i\|_{p,\infty} \le \|T_i\|_{p,q} \le d_i^{\frac{1}{q}} \|T_i\|_{p,\infty} \le \overset{\frac{1}{q}}{\max}(\boldsymbol{d}_{-1}) \|T_i\|_{p,\infty}$$

for $i = 1, 2, \cdots, k$, and $\|T_{k+1}\|_{p,q} \le d_{k+1}^{\frac{1}{q}} \|T_{k+1}\|_{p,\infty}$. Thus we get the desired result by Part (a).

Regarding Part (d), we first show the result holds when $k_1 = k_2$. For any $g \in \mathcal{N}_{p,q,c,c_o}^{k_1,\boldsymbol{d}^1}$, we could add $d_i^2 - d_i^1$ neurons in each hidden layer with no connection to other neurons, thus not increasing the norm of each layer. Note that this neural network belongs to $\mathcal{N}_{p,q,c,c_o}^{k_1,\boldsymbol{d}^2}$.

For the general case when $k_1 \le k_2$, we could add $k_2 - k_1$ identity layers of width 1 with their $L_{p,q}$ norm equals $1 \le c$. Then the new neural network represents the same function as the original one. Combining the conclusion of Part (a), we have

$$\mathcal{N}_{p,q,c,c_o}^{k_1,\boldsymbol{d}^1} \subseteq \mathcal{N}_{p,q,c,c_o}^{k_2,\check{\boldsymbol{d}}^1},$$

where $\tilde{d}_i^1 = d_i^1$ for $i = 0, 1, \cdots, k_1$, and $\tilde{d}_i^1 = d_{k_1+1}^1$ for $i = k_1 + 1, \cdots, k_2 + 1$. Note that $\mathcal{N}_{p,q,c,c_o}^{k_2,\check{\boldsymbol{d}}^1} \subseteq \mathcal{N}_{p,q,c,c_o}^{k_2,\boldsymbol{d}^2}$ by the case when $k_1 = k_2$. Thus we get what is expected. $\qquad\square$

## C  Radermacher Complexities

Rademacher complexity is commonly used to measure the complexity of a hypothesis class with respect to a probability distribution or a sample and analyze generalization bounds [6].

**Rademacher Complexities.**  The *empirical Rademacher complexity* of the hypothesis class $\mathcal{F}$ with respect to a data set $S = \{z_1 \ldots z_n\}$ is defined as:

$$\widehat{\mathfrak{R}}_S(\mathcal{F}) = \mathbb{E}_\epsilon \left[ \sup_{f \in \mathcal{F}} \left( \frac{1}{n} \sum_{i=1}^n \epsilon_i f(z_i) \right) \right]$$

where $\epsilon = \{\epsilon_1 \dots \epsilon_n\}$ are $n$ independent Rademacher random variables. The *Rademacher complexity* of the hypothesis class $\mathcal{F}$ with respect to $n$ samples is defined as:

$$\mathfrak{R}_n(\mathcal{F}) = \mathbb{E}_{S \sim \mathcal{D}^n} \left[ \widehat{\mathfrak{R}}_S(\mathcal{F}) \right]$$

We list the following technical lemmas that will be used later in our own proofs for reference.

**Lemma 2.** *Let $\mathcal{F}$ and $\mathcal{G}$ be two hypothesis classes and $a \in \mathbb{R}$ be a constant. Define the shorthand notation:*

$$a\mathcal{F} = \{af \mid f \in \mathcal{F}\}$$
$$\mathcal{F} + \mathcal{G} = \{f + g \mid f \in \mathcal{F} \text{ and } g \in \mathcal{G}\}$$

*We have:*

i. $\widehat{\mathfrak{R}}_S(a\mathcal{F}) = |a| \, \widehat{\mathfrak{R}}_S(\mathcal{F})$

ii. $\mathcal{F} \subseteq \mathcal{G} \;\Rightarrow\; \widehat{\mathfrak{R}}_S(\mathcal{F}) \leq \widehat{\mathfrak{R}}_S(\mathcal{G})$

iii. $\widehat{\mathfrak{R}}_S(\mathcal{F} + \mathcal{G}) \leq \widehat{\mathfrak{R}}_S(\mathcal{F}) + \widehat{\mathfrak{R}}_S(\mathcal{G})$

*Proof.* By definition. $\square$

**Lemma 3.** *[15] Assume that the hypothesis class $\mathcal{F} \subseteq \{f | f : \mathcal{X} \to \mathbb{R}\}$ and $\boldsymbol{x}_1, \cdots, \boldsymbol{x}_n \in \mathcal{X}$. Let $G : \mathbb{R} \to \mathbb{R}$ be convex and increasing. Assume that the function $\phi : \mathbb{R} \to \mathbb{R}$ is $L$-Lipschitz continuous and satisfies that $\phi(0) = 0$. We have:*

$$\mathbb{E}_\epsilon \left[ G \left( \sup_{f \in \mathcal{F}} \left( \frac{1}{n} \sum_{i=1}^n \epsilon_i \phi(f(\boldsymbol{x}_i)) \right) \right) \right] \leq \mathbb{E}_\epsilon \left[ G \left( L \sup_{f \in \mathcal{F}} \left( \frac{1}{n} \sum_{i=1}^n \epsilon_i f(\boldsymbol{x}_i) \right) \right) \right]$$

**Lemma 4** (Massart's finite lemma)**.** *Let $\mathcal{A}$ be some finite subset of $\mathbb{R}^m$ and $\epsilon_1, \epsilon_2, \cdots, \epsilon_m$ be independent Radermacher random variables. Let $r = \sup_{\boldsymbol{a} \in \mathcal{A}} \|\boldsymbol{a}\|_2$, then we have*

$$\mathbb{E} \left[ \sup_{\boldsymbol{a} \in \mathcal{A}} \frac{1}{m} \sum_{i=1}^m \epsilon_i a_i \right] = \frac{r \sqrt{2 \log |\mathcal{A}|}}{m}$$

The theorem below is a more general version of [17, Theorem 3.1], where they assume $a = 0$, of which the proof is very similar to the original one.

**Theorem 3.** *Let $z$ be a random variable of support $\mathcal{Z}$ and distribution $\mathcal{D}$. Let $S = \{z_1 \dots z_n\}$ be a data set of $n$ i.i.d. samples drawn from $\mathcal{D}$. Let $\mathcal{F}$ be a hypothesis class satisfying $\mathcal{F} \subseteq \{f \mid f : \mathcal{Z} \to [a, a+1]\}$. Fix $\delta \in (0,1)$. With probability at least $1 - \delta$ over the choice of $S$, the following holds for all $h \in \mathcal{F}$:*

$$\mathbb{E}_\mathcal{D}[h] \leq \widehat{\mathbb{E}}_S[h] + 2\mathfrak{R}_n(\mathcal{F}) + \sqrt{\frac{\log(1/\delta)}{2n}}$$

## D  Propositions 1, 2, 3

In this section, define $\sigma(u) = uI\{u > 0\}$ for $u \in \mathbb{R}$ and $\sigma \circ \mathbf{z} = (\sigma(z_1), \cdots, \sigma(z_m))$ for any vector $\mathbf{z} \in \mathbb{R}^m$.

### D.1  Proof for Proposition 1

*Proof.* By Theorem 1, $\mathcal{N}_{1,q,c,c_o}^{k,\boldsymbol{d}} \subseteq \mathcal{N}_{1,\infty,c,c_o}^{k,\boldsymbol{d}}$. Therefore it is sufficient to show that the result holds for $\mathcal{N}_{1,\infty,c,c_o}^{k,\boldsymbol{d}}$.

In order to get the first term inside the minimum operator, we will show that $\mathcal{N}_{1,\infty,c,c_o}^{k,\boldsymbol{d}}$ belongs to some DNN class with only bias neuron in the input layer. Then the result follows from Theorem

2[10]. Define $\mathcal{N}_{\gamma_{1,\infty}\leq\gamma}^{k,\boldsymbol{d}^+}$ as a function class that contains all functions representable by $f = T_{k+1} \circ \sigma \circ T_k \circ \cdots \circ \sigma \circ T_1 \circ \boldsymbol{x}$ satisfying that

$$\gamma_{1,\infty} = \prod_{i=1}^{k+1} \|\mathbf{W}_i\|_{1,\infty} \leq \gamma,$$

where $\boldsymbol{d}^+ = (m_1 + 1, d_1 + 1, d_2 + 1, \cdots, d_k + 1, 1)$, $T_i(\mathbf{u}) = \boldsymbol{W}_i^T \mathbf{u}$, and $\boldsymbol{W}_i \in \mathbb{R}^{d_{i-1}^+ \times d_i^+}$ for $i = 1, \cdots, k+1$.

The next step is to prove that $\mathcal{N}_{1,\infty,c,c_o}^{k,\boldsymbol{d}} \subseteq \mathcal{N}_{\gamma_{1,\infty}\leq\max(1,c)^k c_o}^{k,\boldsymbol{d}^+}$. Following the notations in Section 2, for any $\tilde{\mathbf{V}}_i \in \mathbb{R}^{(d_{i-1}+1)\times d_i}$ satisfying that $\left\|\tilde{\mathbf{V}}_i\right\|_{1,\infty} = c$, we have $\|\mathbf{V}_i\|_{1,\infty} = \max(1,c)$, where $\mathbf{V}_i = (\boldsymbol{e}_{1i}, \tilde{\mathbf{V}}_i)$ and $\boldsymbol{e}_{1i} = (1,0,\cdots,0)^T \in \mathbb{R}^{d_{i-1}+1}$. Equivalently, the bias neuron in the $i$th hidden layer can be regarded as a hidden neuron computed from the $i-1$th layer by $\sigma(\boldsymbol{e}_{1i}^T f_{i-1}^*(\boldsymbol{x})) = 1$, while the new affine transformation could be parameterized by $\mathbf{V}_i$, such that $\|\mathbf{V}_i\|_{1,\infty} = \max(1,c)$.

Finally, we get the first term inside the minimum operator by applying Theorem 2[10], and the second term is the bound of Proposition 2 when $p = 1$. $\square$

### D.2 Proposition 2

We first introduce two technical lemmas, which will be used later to prove Proposition 2.

**Lemma 5.** $\mathbf{z}_i \in \mathbb{R}^{m_1}, \|\mathbf{z}_i\|_\infty \leq 1$ for $i = 1, 2, \cdots, n$. For $p \in (1, 2]$,

$$\frac{1}{n}\mathbb{E}\left\|\sum_{i=1}^{n}\epsilon_i\mathbf{z}_i\right\|_{p^*} \leq \frac{m_1^{\frac{1}{p^*}}}{\sqrt{n}}\min\left((\sqrt{p^*-1}, \sqrt{2\log(2m_1)})\right),$$

and for $p = 1 \cup (2,\infty)$,

$$\frac{1}{n}\mathbb{E}\left\|\sum_{i=1}^{n}\epsilon_i\mathbf{z}_i\right\|_{p^*} \leq \sqrt{\frac{2\log(2m_1)}{n}}m_1^{\frac{1}{p^*}}.$$

**Lemma 6.** $\forall p, q \geq 1$, $s_1, s_2 \geq 1$, $\epsilon \in \{-1, +1\}^n$ and for all functions $g : \mathbb{R}^{m_1} \to \mathbb{R}^{s_1}$, we have

$$\sup_{\mathbf{V}\in\mathbb{R}^{s_1\times s_2}} \frac{1}{\|\mathbf{V}\|_{p,q}}\left\|\sum_{i=1}^{n}\epsilon_i\sigma\circ\left(\mathbf{V}^T g(\boldsymbol{x}_i)\right)\right\|_{p^*} = s_2^{[\frac{1}{p^*}-\frac{1}{q}]_+}\sup_{\boldsymbol{v}\in\mathbb{R}^{s_1}}\frac{1}{\|\boldsymbol{v}\|_p}\left|\sum_{i=1}^{n}\epsilon_i\sigma\left(\langle\boldsymbol{v}, g(\boldsymbol{x}_i)\rangle\right)\right|,$$

where $\frac{1}{p} + \frac{1}{p^*} = 1$.

### D.3 Proof of Proposition 2

*Proof.* The proof has two main steps.

Fixing the sample $S$, $p \geq 1$ and the architecture of the DNN, define a series of random variables $\{Z_0, Z_1, \cdots, Z_k\}$ as

$$Z_0 = \left\|\sum_{i=1}^{n}\epsilon_i\boldsymbol{x}_i\right\|_{p^*}$$

and

$$Z_j = \sup_{f\in\mathcal{N}_{p,q,c,c_o}^{k,\boldsymbol{d}}}\left\|\sum_{i=1}^{n}\epsilon_i\sigma\circ f_j(\boldsymbol{x}_i)\right\|_{p^*},$$

for $j = 1, \cdots, k$, where $\{\epsilon_1, \cdots, \epsilon_n\}$ are $n$ independent Rademacher random variables, and $f_j$ denotes the $j$th hidden layer of the WN-DNN $f$.

In the first step, we prove by induction that for $j = 1, \cdots, k$ and any $t \in \mathbb{R}$

$$\mathbb{E}_\epsilon \exp(tZ_j) \leq 4^j \exp\left(\frac{t^2 ns_j^2}{2} + tc^j\prod_{i=1}^{j}d_i^{[\frac{1}{p^*}-\frac{1}{q}]_+}A_{m_1,S}^p\right),$$

where
$$s_j = \sum_{i=2}^{j} c^{j-i+1} \prod_{l=i}^{j} d_l^{[\frac{1}{p^*} - \frac{1}{q}]_+} + (m_1^{\frac{1}{p^*}} + 1)c^j \prod_{l=1}^{j} d_l^{[\frac{1}{p^*} - \frac{1}{q}]_+}$$

and

$$A_{m_1,S}^p = \begin{cases} \sqrt{n} \min\left((\sqrt{p^*-1}m_1^{\frac{1}{p^*}}, \sqrt{2\log(2m_1)}m_1^{\frac{1}{p^*}}\right) & \text{if} \quad p \in (1,2] \\ \sqrt{2n\log(2m_1)}m_1^{\frac{1}{p^*}} & \text{if} \quad p \in 1 \cup (2,\infty) \end{cases}$$

Note that $s_{j+1} = cd_j^{[\frac{1}{p^*} - \frac{1}{q}]_+}(s_j + 1)$.

When $j = 0$, by Lemma 5, $E_\epsilon Z_0 \leq A_{m_1,S}^p$. Note that $Z_0$ is a deterministic function of the i.i.d.random variables $\epsilon_1, \cdots, \epsilon_n$, satisfying that

$$|Z_0(\epsilon_1, \cdots, \epsilon_i, \cdots, \epsilon_n) - Z_0(\epsilon_1, \cdots, -\epsilon_i, \cdots, \epsilon_n)| \leq 2\max\|\mathbf{x}_i\|_{p^*} \leq 2m_1^{\frac{1}{p^*}}$$

by Minkowski inequality. By the proof of Theorem 6.2 [7], $Z_0$ satisfies that $\log \mathbb{E}_\epsilon \exp\left(t(Z_0 - E_\epsilon Z_0)\right) \leq t^2 nm_1^{\frac{2}{p^*}}/2$, thus

$$\mathbb{E}_\epsilon \exp\left(tZ_0\right) = \mathbb{E}_\epsilon \exp\left(t(Z_0 - E_\epsilon Z_0)\right) * \exp\left(tE_\epsilon Z_0\right)$$

$$\leq \exp\left(\frac{t^2 nm_1^{\frac{2}{p^*}}}{2} + tA_{m_1,S}^p\right)$$

for any $t \in \mathbb{R}$.

For the case when $j = 1, \cdots, k$,

$$\mathbb{E}_\epsilon \exp\left(tZ_j\right) = \mathbb{E}_\epsilon \exp\left(t \sup_{\|\tilde{\mathbf{V}}_j\|_{p,q} \leq c} \left\|\sum_{i=1}^{n} \epsilon_i \sigma \circ \left(\tilde{\mathbf{V}}_j^T \sigma \circ f_{j-1}^*(\boldsymbol{x}_i)\right)\right\|_{p^*}\right)$$

$$= \mathbb{E}_\epsilon \exp\left(tcd_j^{[\frac{1}{p^*} - \frac{1}{q}]_+} \sup_{\boldsymbol{v},f} \left|\sum_{i=1}^{n} \epsilon_i \sigma(\boldsymbol{v}^T \sigma \circ f_{j-1}^*(\boldsymbol{x}_i))/\|\boldsymbol{v}\|_p\right|\right) \tag{8a}$$

$$\leq 2\mathbb{E}_\epsilon \exp\left(tcd_j^{[\frac{1}{p^*} - \frac{1}{q}]_+} \sup_{\boldsymbol{v},f} \sum_{i=1}^{n} \epsilon_i \sigma(\boldsymbol{v}^T \sigma \circ f_{j-1}^*(\boldsymbol{x}_i))/\|\boldsymbol{v}\|_p\right) \tag{8b}$$

$$\leq 2\mathbb{E}_\epsilon \exp\left(tcd_j^{[\frac{1}{p^*} - \frac{1}{q}]_+} \sup_{\boldsymbol{v},f} \boldsymbol{v}^T \sum_{i=1}^{n} \epsilon_i \sigma \circ f_{j-1}^*(\boldsymbol{x}_i)/\|\boldsymbol{v}\|_p\right) \tag{8c}$$

$$\leq 2\mathbb{E}_\epsilon \exp\left(tcd_j^{[\frac{1}{p^*} - \frac{1}{q}]_+} \sup_f \left\|\sum_{i=1}^{n} \epsilon_i(1, \sigma \circ f_{j-1}(\boldsymbol{x}_i))\right\|_{p^*}\right)$$

$$\leq 2\mathbb{E}_\epsilon \exp\left(tcd_j^{[\frac{1}{p^*} - \frac{1}{q}]_+} (|\sum_{i=1}^{n} \epsilon_i| + \sup_f \left\|\sum_{i=1}^{n} \epsilon_i \sigma \circ f_{j-1}(\boldsymbol{x}_i)\right\|_{p^*})\right)$$

$$\leq 2\left[\mathbb{E}_\epsilon \exp\left(r_j tcd_j^{[\frac{1}{p^*} - \frac{1}{q}]_+} |\sum_{i=1}^{n} \epsilon_i|\right)\right]^{\frac{1}{r_j}} \left[\mathbb{E}_\epsilon \exp\left(r_j^* tcd_j^{[\frac{1}{p^*} - \frac{1}{q}]_+} \sup_f \left\|\sum_{i=1}^{n} \epsilon_i \sigma \circ f_{j-1}(\boldsymbol{x}_i)\right\|_{p^*}\right)\right]^{\frac{1}{r_j^*}}$$

$$\tag{8d}$$

$$\leq 2\left[2\mathbb{E}_\epsilon \exp\left(r_j tcd_j^{[\frac{1}{p^*} - \frac{1}{q}]_+} \sum_{i=1}^{n} \epsilon_i\right)\right]^{\frac{1}{r_j}} \left[\mathbb{E}_\epsilon \exp\left(r_j^* tcd_j^{[\frac{1}{p^*} - \frac{1}{q}]_+} Z_{j-1}\right)\right]^{\frac{1}{r_j^*}}, \tag{8e}$$

$$\leq 4^{1+\frac{j-1}{r_j^*}} \exp\left(\frac{nt^2 c^2 d_j^{2[\frac{1}{p^*} - \frac{1}{q}]_+}(1 + s_{j-1})^2}{2} + tc^j \prod_{i=1}^{j} d_i^{[\frac{1}{p^*} - \frac{1}{q}]_+} A_{m_1,S}^p\right)$$

$$\leq 4^j \exp\left( \frac{nt^2 s_j^2}{2} + t c^j \prod_{i=1}^{j} d_i^{[\frac{1}{p^*} - \frac{1}{q}]_+} A_{m_1,S}^p \right)$$

The step in Equation (8a) follows from Lemma 6. The step in Equation (8b) follows from the observation that

$$\mathbb{E}_\epsilon \exp\left( \sup_{\boldsymbol{v}} \left| \sum_{i=1}^{n} \epsilon_i \frac{\sigma(\boldsymbol{v}^T f_{j-1}^*(\boldsymbol{x}_i))}{\|\boldsymbol{v}\|_p} \right| \right) \leq \mathbb{E}_\epsilon \exp\left( \sup_{\boldsymbol{v}} \sum_{i=1}^{n} \epsilon_i \frac{\sigma(\boldsymbol{v}^T f_{j-1}^*(\boldsymbol{x}_i))}{\|\boldsymbol{v}\|_p} \right) +$$

$$\mathbb{E}_\epsilon \exp\left( \sup_{\boldsymbol{v}} \sum_{i=1}^{n} (-\epsilon_i) \frac{\sigma(\boldsymbol{v}^T f_{j-1}^*(\boldsymbol{x}_i))}{\|\boldsymbol{v}\|_p} \right) = 2\mathbb{E}_\epsilon \exp\left( \sup_{\boldsymbol{v}} \sum_{i=1}^{n} \epsilon_i \frac{\sigma(\boldsymbol{v}^T f_{j-1}^*(\boldsymbol{x}_i))}{\|\boldsymbol{v}\|_p} \right).$$

The step in Equation (8c) follows from Lemma 3. Note that Equation (8d) holds for any $r > 1$ and $r^* = \frac{r}{r-1}$ by Hölder's inequality $\mathbb{E}(|XY|) \leq \mathbb{E}(|X|^r)^{\frac{1}{r}} \mathbb{E}(|Y|^{r^*})^{\frac{1}{r^*}}$. An optimal $r_j = s_{j-1} + 1$ is chosen in our case. The step in Equation (8e) follows from $\mathbb{E}_\epsilon \exp(|X|) \leq \mathbb{E}_\epsilon \exp(X) + \mathbb{E}_\epsilon \exp(-X)$.

Note that $\sum_{i=1}^{n} \epsilon_i$ is also a deterministic function of the i.i.d. random variables $\epsilon_1, \cdots, \epsilon_n$, satisfying that $\mathbb{E}_\epsilon \sum_{i=1}^{n} \epsilon_i = 0$ and

$$\left| \sum_{i \neq j} \epsilon_i + \epsilon_j - \left( \sum_{i \neq j} \epsilon_i - \epsilon_j \right) \right| \leq 2.$$

Then by the proof of Theorem 6.2 [7],

$$\mathbb{E}_\epsilon \exp\left( t \sum_{i=1}^{n} \epsilon_i \right) \leq \exp\left( \frac{t^2 n}{2} \right)$$

for any $t \in \mathbb{R}$. Then we get the desired result by choosing the optimal $r_j$ while following the induction assumption.

The second step is based on the idea of [10] using Jensen's inequality. For any $\lambda > 0$,

$$n\widehat{\mathfrak{R}}_S(\mathcal{N}_{p,q,c,c_o}^{k,\boldsymbol{d}}) = \mathbb{E}_\epsilon \left[ \sup_{f \in \mathcal{N}_{p,q,c,c_o}^{k,\boldsymbol{d}}} \left( \sum_{i=1}^{n} \epsilon_i f(\boldsymbol{x}_i) \right) \right]$$

$$\leq \frac{1}{\lambda} \log \mathbb{E}_\epsilon \exp\left( \lambda \sup_{f \in \mathcal{N}_{p,q,c,c_o}^{k,\boldsymbol{d}}} \left( \sum_{i=1}^{n} \epsilon_i f(\boldsymbol{x}_i) \right) \right)$$

$$\leq \frac{1}{\lambda} \log \mathbb{E}_\epsilon \exp\left( \lambda c_o \sup_{f \in \mathcal{N}_{p,q,c,c_o}^{k,\boldsymbol{d}}} \left\| \sum_{i=1}^{n} \epsilon_i (1, \sigma \circ f_k(\boldsymbol{x}_i)) \right\|_{p^*} \right)$$

$$\leq \frac{1}{\lambda} \left[ (k+1) \log 4 + \frac{\lambda^2 c_o^2 n (s_k + 1)^2}{2} + \lambda A_{m_1,S}^p c_o c^k \prod_{i=1}^{k} d_i^{[\frac{1}{p^*} - \frac{1}{q}]_+} \right] \qquad (9a)$$

$$= \frac{(k+1) \log 4}{\lambda} + \frac{\lambda c_o^2 n (s_k + 1)^2}{2} + c_o c^k \prod_{i=1}^{k} d_i^{[\frac{1}{p^*} - \frac{1}{q}]_+} A_{m_1,S}^p,$$

where the step in Equation (9a) is derived using a similar techinique as in Equations (8a) to (8e) By choosing the optimal $\lambda = \frac{\sqrt{(k+1) \log 16}}{c_o (s_k + 1) \sqrt{n}}$, we have

$$\widehat{\mathfrak{R}}_S(\mathcal{N}_{p,q,c}^{k,\boldsymbol{d}}) \leq c_o \sqrt{\frac{(k+1) \log 16}{n}} \left( \sum_{i=2}^{k} c^{k-i+1} \prod_{l=i}^{k} d_l^{[\frac{1}{p^*} - \frac{1}{q}]_+} + (m_1^{\frac{1}{p^*}} + 1) c^k \prod_{i=1}^{k} d_i^{[\frac{1}{p^*} - \frac{1}{q}]_+} + 1 \right) +$$

$$\frac{1}{\sqrt{n}} c_o c^k \prod_{i=1}^{k} d_i^{[\frac{1}{p^*} - \frac{1}{q}]_+} A_{m_1,S}^p$$

$\square$

### D.4 Proof of Lemma 5

*Proof.* For $p \in (1, 2]$, or equivalently $p^* \in [2, \infty)$, $\|\cdot\|_{p^*}$ is $2(p^* - 1)$-strongly convex with respect to itself on $\mathbb{R}^{m_1+1}$ [24] and $\|\mathbf{z}_i\|_{p^*} \le m_1^{\frac{1}{p^*}} \|\mathbf{z}_i\|_\infty$, thus $\frac{1}{n}\mathbb{E}\left\|\sum_{i=1}^n \epsilon_i \mathbf{z}_i\right\|_{p^*} \le \sqrt{\frac{p^*-1}{n}} m_1^{\frac{1}{p^*}}$ [14].

For $p \in [1, \infty)$ or equivalently $p^* \in (1, \infty]$, let $\mathbf{z}[j] = (\mathbf{z}_1[j], \mathbf{z}_2[j], \cdots, \mathbf{z}_n[j])^T$, where $\mathbf{z}_i[j]$ is the $j$th element of the vector $\mathbf{z}_i \in \mathbb{R}^{m_1}$.

$$
\begin{aligned}
\frac{1}{n}\mathbb{E}\left\|\sum_{i=1}^n \epsilon_i \mathbf{z}_i\right\|_{p^*} &\le \frac{m_1^{\frac{1}{p^*}}}{n}\mathbb{E}\left\|\sum_{i=1}^n \epsilon_i \mathbf{z}_i\right\|_\infty \\
&\le \frac{m_1^{\frac{1}{p^*}} \sqrt{2\log(2m_1)}}{n} \sup_j \|\mathbf{z}[j]\|_2 \\
&\le \frac{m_1^{\frac{1}{p^*}} \sqrt{2\log(2m_1)}}{n} \sqrt{n} \sup_j \|\mathbf{z}[j]\|_\infty \\
&\le \frac{m_1^{\frac{1}{p^*}} \sqrt{2\log(2m_1)}}{\sqrt{n}}
\end{aligned}
$$
(10)

The step in Equation (10) follows from Lemma 4. $\qquad\square$

### D.5 Proof of Lemma 6

*Proof.* The proof is based on the ideas of [19, Lemma 17]

The right hand side (RHS) is always less than or equal to the left hand side (LHS), since given any vector $\boldsymbol{v}$ we could create a corresponding matrix $\mathbf{V}$ of which each row is $\boldsymbol{v}$.

Then we will show that (LHS) is always less than or equal to (RHS). Let $\mathbf{V}[, j]$ be the $j$th column of the matrix $\mathbf{V}$. We have $\|\mathbf{V}\|_{p,p^*} \le \|\mathbf{V}\|_{p,q}$ when $q \le p^*$ and by Hölder's inequality, $\|\mathbf{V}\|_{p,p^*} \le s_2^{[\frac{1}{p^*}-\frac{1}{q}]}\|\mathbf{V}\|_{p,q}$ when $q > p^*$. Thus

$$
\begin{aligned}
(\text{LHS}) &\le \sup_{\mathbf{V}\in\mathbb{R}^{s_1\times s_2}} \frac{s_2^{[\frac{1}{p^*}-\frac{1}{q}]_+}}{\|\mathbf{V}\|_{p,p^*}}\left\|\sum_{i=1}^n \epsilon_i \sigma \circ \left(\mathbf{V}^T g(\boldsymbol{x}_i)\right)\right\|_{p^*} \\
&= s_2^{[\frac{1}{p^*}-\frac{1}{q}]_+} \sup_{\mathbf{V}\in\mathbb{R}^{s_1\times s_2}} \frac{1}{\|\mathbf{V}\|_{p,p^*}}\left(\sum_{j=1}^{s_2}\left|\sum_{i=1}^n \epsilon_i \sigma\left(\langle\mathbf{V}[, j], g(\boldsymbol{x}_i)\rangle\right)\right|^{p^*}\right)^{1/p^*} \\
&\le s_2^{[\frac{1}{p^*}-\frac{1}{q}]_+} \sup_{\mathbf{V}\in\mathbb{R}^{s_1\times s_2}} \frac{1}{\|\mathbf{V}\|_{p,p^*}}\left(\sum_{j=1}^{s_2}\left(\|\mathbf{V}[, j]\|_p \frac{(\text{RHS})}{s_2^{[\frac{1}{p^*}-\frac{1}{q}]_+}}\right)^{p^*}\right)^{1/p^*} \\
&= (\text{RHS}) \sup_{\mathbf{V}\in\mathbb{R}^{s_1\times s_2}} \frac{1}{\|\mathbf{V}\|_{p,p^*}}\left(\sum_{j=1}^{s_2}(\|\mathbf{V}[, j]\|_p)^{p^*}\right)^{1/p^*} \\
&= (\text{RHS})
\end{aligned}
$$

$\square$

### D.6 Proposition 3

*Proof.* Define $\mathcal{N}_{\gamma_{p,q}\le\gamma}^{k,\boldsymbol{d}}$ as a function class that contains all functions representable by some neural network $f = T_{k+1} \circ \sigma \circ T_k \circ \cdots \circ \sigma \circ T_1 \circ \boldsymbol{x}$ satisfying that

$$
\gamma_{p,q} = \prod_{i=1}^{k+1} \|\mathbf{W}_i\|_{p,q} \le \gamma,
$$

where $\boldsymbol{d} = (m_1, d, \cdots, d, 1)$, $T_i(\mathbf{u}) = \boldsymbol{W}_i^T \mathbf{u}$, and $\boldsymbol{W}_i \in \mathbb{R}^{d_{i-1} \times d_i}$ for $i = 1, \cdots, k+1$. In order to use the conclusion of [19, Theorem 3] for DNNs with no bias neuron, it is sufficient to show that

$$\mathcal{N}_{\gamma_{p,q} \leq \gamma}^{k,\boldsymbol{d}} \subseteq \mathcal{N}_{p,q,c,c_o}^{k,\boldsymbol{d}},$$

for any $c, c_o$ satisfying that $c^k c_o \geq \gamma$.

If any $\|T_i\|_{p,q} = 0$, then $f = 0 \in \mathcal{N}_{p,q,c,c_o}^{k,\boldsymbol{d}}$. Otherwise, for any $c, c_o$ satisfying that $c^k c_o \geq \gamma \geq \prod_{i=1}^{k+1} \|T_i\|_{p,q}$, we rescale each hidden layer by

$$s_i = c/ \|T_i\|_{p,q},$$

that is, define $T_i^*$ by $\mathbf{B}_i^* = 0$ and $\mathbf{W}_i^* = s_i \mathbf{W}_i$, such that $\|T_i^*\|_{p,q} = c$ and $T_i^* = s_i T_i$. Correspondingly, rescale the output layer by $1/ \prod_{i=1}^{k} s_i$ and $\|T_{k+1}^*\|_{p,q} \leq c_o$ as $s_i \geq 1$. Therefore, $f \in \mathcal{N}_{p,q,c,c_o}^{k,\boldsymbol{d}}$. $\qquad\qquad\qquad\square$

# E    Generalization Bounds

In this section, we provide a generalization bound that holds for any data distribution for regression as an extension of Section 3.

**The Regression Problem.**    Assume that $(\boldsymbol{x}_1, y_1), \ldots, (\boldsymbol{x}_n, y_n)$ are $n$ i.i.d samples on $\mathcal{X} \times \mathcal{Y} \subseteq \mathbb{R}^{m_1} \times \mathbb{R}$, satisfying that

$$y_i = f(\boldsymbol{x}_i) + \varepsilon_i, \tag{11}$$

where $f : \mathcal{X} \to \mathcal{Y} \subseteq \mathbb{R}$ is an unknown function and $\varepsilon_i$ an independent noise.

## E.1    Generalization Bounds

Assume that $d : \mathcal{Y} \times \mathcal{Y} \to [0, 1]$ is a 1-Lipschitz function related to the prediction problem. For example, we could define $d(y, y') = \min(1, (y - y')^2/2)$. Let $\mathbf{z} = (\boldsymbol{x}, y) \in \mathcal{Z}$, where $\mathcal{Z} = \mathcal{X} \times \mathcal{Y}$. Furthermore, for each $f \in \mathcal{N}_{p,q,c,c_o}^{k,\boldsymbol{d}}$, define a corresponding $h_f$ such that $h_f(\mathbf{z}) = d(y, f(\boldsymbol{x}))$. Let $\mathcal{H}_{p,q,c,c_o}^{k,\boldsymbol{d}}$ be a hypothesis class satisfying

$$\mathcal{H}_{p,q,c,c_o}^{k,\boldsymbol{d}} = \bigcup_{f \in \mathcal{N}_{p,q,c,c_o}^{k,\boldsymbol{d}}} h_f.$$

For every $h \in \mathcal{H}_{p,q,c,c_o}^{k,\boldsymbol{d}}$, define the true and empirical risks as

$$\mathbb{E}_{\mathcal{D}}[h] = \mathbb{E}_{\mathbf{z} \sim \mathcal{D}}[h(\mathbf{z})], \quad \widehat{\mathbb{E}}_S[h] = \frac{1}{n} \sum_{i=1}^{n} h(\mathbf{z}_i).$$

**Theorem 4.** *Let $\mathbf{z} = (\boldsymbol{x}, y)$ be a random variable of support $\mathcal{Z}$ and distribution $\mathcal{D}$. Let $S = \{\mathbf{z}_1 \ldots \mathbf{z}_n\}$ be a dataset of $n$ i.i.d. samples drawn from $\mathcal{D}$. Fix $\delta \in (0, 1)$, $k \in [0, \infty)$ and $d_i \in \mathbb{N}_+$ for $i = 1, \cdots, k$. With probability at least $1 - \delta$ over the choice of $S$,*

*(a) for $p = 1$ and $q \in [1, \infty]$, we have $\forall h \in \mathcal{H}_{1,q,c,c_o}^{k,\boldsymbol{d}}$:*

$$\mathbb{E}_{\mathcal{D}}[h] \leq \widehat{\mathbb{E}}_S[h] + \sqrt{\frac{\log(1/\delta)}{2n}} + \frac{2c_o}{\sqrt{n}} * \min \Big(2\max(1, c^k)\sqrt{k + 2 + \log(m_1 + 1)},$$

$$\sqrt{(k+1)\log 16} \sum_{i=0}^{k} c^i + c^k(\sqrt{2\log(2m_1)} + \sqrt{(k+1)\log 16})\Big)$$

(b) *for $p \in (1,2]$ and $q \in [1,\infty]$, we have $\forall h \in \mathcal{H}_{p,q,c,c_o}^{k,\boldsymbol{d}}$:*

$$\mathbb{E}_{\mathcal{D}}[h] \leq \widehat{\mathbb{E}}_S[h] + \sqrt{\frac{\log(1/\delta)}{2n}} + \frac{1}{\sqrt{n}} c_o c^k \prod_{i=1}^{k} d_i^{[\frac{1}{p^*}-\frac{1}{q}]_+} \sqrt{2\log(2m_1)} m_1^{\frac{1}{p^*}} +$$

$$c_o \sqrt{\frac{(k+1)\log 16}{n}} \left( \sum_{i=1}^{k+1} c^{k-i+1} \prod_{l=i}^{k} d_l^{[\frac{1}{p^*}-\frac{1}{q}]_+} + m_1^{\frac{1}{p^*}} c^k \prod_{i=1}^{k} d_i^{[\frac{1}{p^*}-\frac{1}{q}]_+} \right).$$

(c) *for $p \in (2,\infty)$ and $q \in [1,\infty]$, we have $\forall h \in \mathcal{H}_{p,q,c,c_o}^{k,\boldsymbol{d}}$:*

$$\mathbb{E}_{\mathcal{D}}[h] \leq \widehat{\mathbb{E}}_S[h] + \sqrt{\frac{\log(1/\delta)}{2n}} +$$

$$\frac{1}{\sqrt{n}} c_o c^k \prod_{i=1}^{k} d_i^{[\frac{1}{p^*}-\frac{1}{q}]_+} m_1^{\frac{1}{p^*}} \min\left( (\sqrt{p^*-1}, \sqrt{2\log(2m_1)}) \right) +$$

$$c_o \sqrt{\frac{(k+1)\log 16}{n}} \left( \sum_{i=1}^{k+1} c^{k-i+1} \prod_{l=i}^{k} d_l^{[\frac{1}{p^*}-\frac{1}{q}]_+} + m_1^{\frac{1}{p^*}} c^k \prod_{i=1}^{k} d_i^{[\frac{1}{p^*}-\frac{1}{q}]_+} \right).$$

The corollary below gives a generalization bound for the $L_{1,\infty}$ WN-DNNs.

**Corollary 1.** *Let $\mathbf{z} = (\boldsymbol{x}, y)$ be a random variable of support $\mathcal{Z}$ and distribution $\mathcal{D}$. Let $S = \{\mathbf{z}_1 \ldots \mathbf{z}_n\}$ be a dataset of $n$ i.i.d. samples drawn from $\mathcal{D}$. Fix $\delta \in (0,1)$, $k \in [0,\infty)$ and $d_i \in \mathbb{N}_+$ for $i = 1, \cdots, k$. Assume that $c^k \leq a_0$ for some $a_0 \geq 1$. With probability at least $1 - \delta$ over the choice of S, for any $h \in \mathcal{H}_{p,q,c,c_o}^{k,\boldsymbol{d}}$, we have:*

$$\mathbb{E}_{\mathcal{D}}[h] \leq \widehat{\mathbb{E}}_S[h] + \sqrt{\frac{\log(1/\delta)}{2n}} + \frac{4c_o a_0}{\sqrt{n}} \sqrt{k + 2 + \log(m_1 + 1)}.$$

For instance, we could define $c$ as $1 + \frac{v_0}{k}$ with some constant $v_0 \geq 0$ for ResNet [12], then $c(k)^k \leq e^{v_0}$. The case with $v_0 = 0$ leads to a specific case where the normalization constant $c = 1$.

### E.2 Proof of Theorem 4

*Proof.* By applying Theorem 3, with probability at least $1 - \delta$ over the choice of $S$, $\forall h \in \mathcal{H}_{p,q,c,c_o}^{k,\boldsymbol{d}}$, we have:

$$\mathbb{E}_{\mathcal{D}}[h] - \widehat{\mathbb{E}}_S[h] \leq 2\mathfrak{R}_n(\mathcal{H}_{p,q,c,c_o}^{k,\boldsymbol{d}}) + \sqrt{\frac{\log(1/\delta)}{2n}}.$$

Thus it is equivalent to bound $\mathfrak{R}_n(\mathcal{H}_{p,q,c,c_o}^{k,\boldsymbol{d}})$ in order to bound the absolute value of the generalization error. By Lemma 3, we have:

$$\mathfrak{R}_n(\mathcal{H}_{p,q,c,c_o}^{k,\boldsymbol{d}}) \leq \mathfrak{R}_n(\mathcal{N}_{p,q,c,c_o}^{k,\boldsymbol{d}}).$$

Finally, (a) follows from

$$\mathfrak{R}_n(\mathcal{N}_{p,q,c,c_o}^{k,\boldsymbol{d}}) \leq \sup_S \widehat{\mathfrak{R}}_S(\mathcal{N}_{p,q,c,c_o}^{k,\boldsymbol{d}})$$

and Proposition 1, while

$$\mathfrak{R}_n(\mathcal{N}_{p,q,c,c_o}^{k,\boldsymbol{d}}) = \mathfrak{R}_n(\mathcal{N}_{p,q,c,c_o}^{k,\boldsymbol{d}}) \leq \sup_S \widehat{\mathfrak{R}}_S(\mathcal{N}_{p,q,c,c_o}^{k,\boldsymbol{d}})$$

and Proposition 2 lead to (b) and (c). $\qquad\square$

## F  Theorem 2

### F.1 Proof of Lemma 1

*Proof.* $\left\| (b_i, \boldsymbol{w}_i^T) \right\|_1 = 1$ implies $\left\| (b_i, 2\boldsymbol{w}_i^T) \right\|_1 \leq 2$, thus by Theorem 1 Part (b), it is sufficient to show that g could be represented by some neural network in $\mathcal{N}_{p,q,wid_k^{1/q},c_o}^{k,\boldsymbol{d}^k}$ if instead $\left\| (b_i, 2\boldsymbol{w}_i^T) \right\|_1 =$

1. In addition, by Theorem 1 Parts (b), (c) and (d), it is equivalent to show that when $\sum_{i=1}^{r}|c_i|=1$, g could be represented by some neural network in $\mathcal{N}_{1,\infty,1,1}^{k,\boldsymbol{d}}$ where $d_i \leq \lceil r/k\rceil + 2m_1 + 3$ for $i = 1,\cdots,k$.

Decompose the shallow neural network as

$$g(\boldsymbol{x}) = \left(\sum_{i=1}^{r_1} c_i^+\right) g_+(\boldsymbol{x}) - \left(\sum_{i=1}^{r_2} c_i^-\right) g_-(\boldsymbol{x}),$$

where

$$g_+(\boldsymbol{x}) = \sum_{i=1}^{r_1} c_i^+ \sigma\left((\boldsymbol{w}_i^+)^T\boldsymbol{x} + b_i^+\right) \Big/ \sum_{i=1}^{r_1} c_i^+, \quad g_-(\boldsymbol{x}) = \sum_{i=1}^{r_2} c_i^- \sigma\left((\boldsymbol{w}_i^-)^T\boldsymbol{x} + b_i^-\right) \Big/ \sum_{i=1}^{r_2} c_i^-$$

for some $c_i^+, c_i^- > 0$. Note that $\left\|\alpha^T A^T\right\|_1 \leq 1$ if $\alpha \in \mathbb{R}^s$ satisfies that $\|\alpha\|_1 \leq 1$, and $A \in \mathbb{R}^{t\times s}$ satisfies that $\|A\|_{1,\infty} \leq 1$. Additionally

$$\sum_{i=1}^{r_1} c_i^+ + \sum_{i=1}^{r_2} c_i^- = \sum_{i=1}^{r} |c_i| = 1.$$

Thus it is sufficient to show that

$$(g_+(\boldsymbol{x}), g_-(\boldsymbol{x}))$$

could be represented by some neural network in $\mathcal{N}_{1,\infty,1,1}^{k,\boldsymbol{d}}$, where each hidden layer contains both $\sigma \circ \boldsymbol{x}$ and $\sigma \circ (-\boldsymbol{x})$, while satisfying that $d_i \leq \lceil r_1/k\rceil + \lceil r_2/k\rceil + 2m_1 + 2$ for $i = 1,\cdots,k$ and $d_{k+1} = 2$.

When $k = 1$, it is trivial.

When $k = 2$, we construct the first hidden layer consisting of $\lceil r_1/2\rceil + \lceil r_2/2\rceil + 2m_1$ hidden neurons:

$$\{(\boldsymbol{w}_i^+)^T\boldsymbol{x} + b_i^+ : i = 1,\cdots,\lceil r_1/2\rceil\}, \{(\boldsymbol{w}_i^-)^T\boldsymbol{x} + b_i^- : i = 1,\cdots,\lceil r_2/2\rceil\}, \boldsymbol{x}, -\boldsymbol{x}.$$

For the second hidden layer, there are $2 + r - (\lceil r_1/2\rceil + \lceil r_2/2\rceil) + 2m_1$ hidden neurons. The first neuron

$$\eta_1 = \sum_{i=1}^{\lceil r_1/2\rceil} c_i^+ \sigma\left((\boldsymbol{w}_i^+)^T\boldsymbol{x} + b_i^+\right) \Big/ \sum_{i=1}^{\lceil r_1/2\rceil} c_i^+,$$

the second neuron

$$\eta_2 = \sum_{i=1}^{\lceil r_2/2\rceil} c_i^- \sigma\left((\boldsymbol{w}_i^-)^T\boldsymbol{x} + b_i^-\right) \Big/ \sum_{i=1}^{\lceil r_2/2\rceil} c_i^-,$$

then follows $\sigma \circ \boldsymbol{x}$, $\sigma \circ (-\boldsymbol{x})$ and the left $r - (\lceil r_1/2\rceil + \lceil r_2/2\rceil)$ hidden neurons

$$\{\eta_i^+ = (\boldsymbol{w}_i^+)^T\sigma \circ \boldsymbol{x} - (\boldsymbol{w}_i^+)^T\sigma \circ (-\boldsymbol{x}) + b_i^+ : i = \lceil r_1/2\rceil + 1,\cdots,r_1\},$$

$$\{\eta_i^- = (\boldsymbol{w}_i^-)^T\sigma \circ \boldsymbol{x} - (\boldsymbol{w}_i^-)^T\sigma \circ (-\boldsymbol{x}) + b_i^- : i = \lceil r_2/2\rceil + 1,\cdots,r_2\}.$$

The output layer only contains two hidden neurons $(g_+, g_-)$, which could be computed respectively by

$$\frac{\sum_{i=1}^{\lceil r_1/2\rceil} c_i^+}{\sum_{i=1}^{r_1} c_i^+}\sigma(\eta_1) + \sum_{i=\lceil r_1/2\rceil+1}^{r_1} \frac{c_i^+}{\sum_{i=1}^{r_1} c_i^+}\sigma(\eta_i^+) \quad \text{and} \quad \frac{\sum_{i=1}^{\lceil r_2/2\rceil} c_i^-}{\sum_{i=1}^{r_2} c_i^-}\sigma(\eta_2) + \sum_{i=\lceil r_2/2\rceil+1}^{r_2} \frac{c_i^-}{\sum_{i=1}^{r_2} c_i^-}\sigma(\eta_i^-).$$

Thus, we find a neural network in $\mathcal{N}_{1,\infty,1,c_o}^{2,\boldsymbol{d}}$ representing $(g_+, g_-)$, where $d_i \leq \lceil r_1/2\rceil + \lceil r_2/2\rceil + 2m_1 + 2$.

When $k = K$, define $r_1^* = (K-1)[r_1/K], r_2^* = (K-1)[r_2/K], r^* = r_1 + r_2$ and

$$g^*(\boldsymbol{x}) = (g_+^*(\boldsymbol{x}), g_-^*(\boldsymbol{x})) = \left( \frac{1}{\sum\limits_{i=1}^{r_1^*} c_i^+} \sum_{i=1}^{r_1^*} c_i^+ \sigma\left((\boldsymbol{w}_i^+)^T \boldsymbol{x} + b_i^+\right), \frac{1}{\sum\limits_{i=1}^{r_2^*} c_i^-} \sum_{i=1}^{r_2^*} c_i^- \sigma\left((\boldsymbol{w}_i^-)^T \boldsymbol{x} + b_i^-\right) \right).$$

By induction assumption, $g^*$ could be represented $h^* \in \mathcal{N}_{1,\infty,1,1}^{K-1,\boldsymbol{d}^*}$, where $d_i^* \leq [r_1^*/(K-1)] + [r_2^*/(K-1)] + 2m_1 + 2$. In order to construct a WN-DNN representing $(g_+, g_-)$, we keep the first $K-1$ hidden layers of $h^*$ and build the $K$th hidden layer based on the output layer of $h^*$. Since the $(K-1)$th hidden layer contains both $\sigma \circ \boldsymbol{x}$ and $\sigma \circ (-\boldsymbol{x})$. Thus except the original two neurons, we could add

$$\{(\boldsymbol{w}_i^+)^T(\sigma \circ \boldsymbol{x} - \sigma \circ (-\boldsymbol{x})) + b_i^+ : i = r_1^* + 1, \cdots, r_1\},$$

$$\{(\boldsymbol{w}_i^-)^T(\sigma \circ \boldsymbol{x} - \sigma \circ (-\boldsymbol{x})) + b_i^- : i = r_2^* + 1, \cdots, r_2\}, \sigma \circ \boldsymbol{x}), \sigma \circ (-\boldsymbol{x})$$

to the $K$th hidden layer. Note that $\left\|(b_i, 2\boldsymbol{w}_i^T)\right\|_1 = 1$, thus we does not increase the $L_{1,\infty}$ norm of the $K$th transformation by adding these neurons.

We finally construct the output layer by

$$\frac{\sum\limits_{i=1}^{r_1^*} c_i^+}{\sum\limits_{i=1}^{r_1} c_i^+} \sigma(g_+^*(\boldsymbol{x})) + \sum_{i=r_1^*+1}^{r_1} \frac{c_i^+}{\sum\limits_{i=1}^{r_1} c_i^+} \sigma\left((\boldsymbol{w}_i^+)^T \boldsymbol{x} + b_i^+\right),$$

$$\frac{\sum\limits_{i=1}^{r_2^*} c_i^-}{\sum\limits_{i=1}^{r_2} c_i^-} \sigma(g_-^*(\boldsymbol{x})) + \sum_{i=r_2^*+1}^{r_2} \frac{c_i^-}{\sum\limits_{i=1}^{r_2} c_i^-} \sigma\left((\boldsymbol{w}_i^-)^T \boldsymbol{x} + b_i^-\right).$$

Thus, we build a neural network in $\mathcal{N}_{1,\infty,1,1}^{K,\boldsymbol{d}}$ representing $(g_+, g_-)$. The width of the $i$th hidden layer $d_i \leq [r_1/K] + [r_2/K] + 2m_1 + 3$. $\qquad\square$

### F.2 Proof for Theorem 2

*Proof.* Assume $f$ is an arbitrary function defined on $\mathbb{R}^{m_1} \to \mathbb{R}$, satisfying that $\|\boldsymbol{x}_1\|_\infty \leq 1$, $\|\boldsymbol{x}_2\|_\infty \leq 1$, $f(\boldsymbol{x}_1) \leq L$ and $|f(\boldsymbol{x}_1) - f(\boldsymbol{x}_2)| \leq L \|\boldsymbol{x}_1 - \boldsymbol{x}_2\|_\infty$. Following [3, Propositions 1 & 6], for $c_o$ greater than a constant depending only on $m_1$, a fixed $\gamma > 0$, , there exists some function $h(\boldsymbol{x}) : \mathbb{R}^{m_1} \to \mathbb{R} = \sum\limits_{i=1}^{r} c_i \sigma(\boldsymbol{w}_i^T \boldsymbol{x} + b_i)$, satisfying that $\sum\limits_{i=1}^{r} |c_i| \leq c_o$, $\left\|(b_i, \boldsymbol{w}_i^T)\right\|_1 = 1$ and $r \leq c_2(m_1) \gamma^{-\frac{2(m_1+1)}{m_1+4}}$, such that

$$\sup_{\|\boldsymbol{x}\|_\infty \leq 1} |f(\boldsymbol{x}) - h(\boldsymbol{x})| \leq c_o \gamma + c_1(m_1) L (\frac{c_o}{L})^{-\frac{2}{m_1+1}} \log \frac{c_o}{L},$$

where $c_1(m_1)$ and $c_2(m_1)$ are some constants depending only on $m_1$.

By taking $\gamma = c_1(m_1)(c_o/L)^{-1-2/(m_1+1)} \log \frac{c_o}{L}$, we have some function $h(\boldsymbol{x}) = \sum\limits_{i=1}^{r} c_i \sigma(\boldsymbol{w}_i^T \boldsymbol{x} + b_i)$, satisfying that $\sum\limits_{i=1}^{r} |c_i| \leq c_o$, $\left\|(b_i, 2\boldsymbol{w}_i^T)\right\|_1 = 1$ and

$$r \leq C_r(m_1)(\log \frac{c_o}{L})^{-2(m_1+1)/(m_1+4)} \left(\frac{c_o}{L}\right)^{2(m_1+3)/(m_1+4)},$$

such that

$$\sup_{\|\boldsymbol{x}\|_\infty \leq 1} |f(\boldsymbol{x}) - h(\boldsymbol{x})| \leq C(m_1) L (\frac{c_o}{L})^{-\frac{2}{m_1+1}} \log \frac{c_o}{L},$$

where $C_r(m_1)$ and $C(m_1)$ denote some constants that depend only on $m_1$.

By Lemma 1, for any integer $k \in [1, r]$, this $h$ could be represented by a neural network in $\mathcal{N}_{p,\infty,1,c_o}^{k,\boldsymbol{d}^k}$, where $\boldsymbol{d}_0^k = m_1$, $\boldsymbol{d}_i^k = [r/k] + 2m_1 + 3$ for $i = 1, \cdots, k$ and $\boldsymbol{d}_{k+1}^k = 1$. $\quad\square$