[Reviews · NeurIPS 2018]

Reviewer 1



This paper essentially seems to address 2 questions (a) Under certain natural weight constraints what is the Rademacher complexity upperbound for nets when we allow for bias vectors in each layer? and (b) How well can such weight constrained nets approximate Lipschitz functions in the sup norm over compact domains? There is a section 4.1 in the paper which is about generalization bounds for nets doing regression. but to me that does not seem to be an essential part of the paper. Let me first say at the outset that the writing of the paper seems extremely bad and many of the crucial steps in the proofs look unfollowable. As it stands this paper is hardly fit to be made public and needs a thorough rewriting! There are few issues that I have with the section on upperbounding Rademacher complexity as has been done between pages 15 to 19, 1. Isnt the entire point of this analysis is just to show that the Srebro-Tomioka-Neyshabur result (http://proceedings.mlr.press/v40/Neyshabur15.pdf) holds even with biases? If that is the entire point then why is this interesting? I am not sure why we should have expected anything much to change about their de-facto exponential dependances on depth with addition of bias! A far more interesting analysis that could have been done and which would have led to a far more interesting paper is if one could have lifted the Rakhlin-Golowich-Shamir analysis of Rademacher complexity (which does not have the exp(depth) dependance!) to nets with biases. I strongly feel thats the research that should have been done! 2. The writing of the proofs here are extremely difficult to follow and the only reason one can follow anything here is because one can often look up the corresponding step in Srebro-Tomioka-Neyshabur! There are just way too many steps here which make no sense to me : like what does "sup_{\x_i}" even mean in the lines below line 350? How did that appear and why is Massart's Lemma applicable for a sup over a continuum? It hardly makes any sense! I have no clue how equation 11(d) follows from equation 11(c). This is similar to Lemma 17 of Srebro-Tomioka-Neyshabur but this definitely needs a rederivation in this context. The steps following equation 12(b) also make little sense to me because they seem to indicate that one is taking \sigma of an innerproduct whereas this \sigma is I believe defined over vectors. Its again not clear as to how the composition structure below equation 12(d) got broken. Now coming to the proof of Theorem 3 - its again fraught with too many problems. Lemma 9 is a very interesting proposition and I wish this proof were clear and convincing. As of now its riddled with too many ambiguous steps! For example : below line 464 what is being computed are \floor(r_1/2) and \floor(r_2/2) number of linear functions apart from x and -x. Any linear function needs 2 ReLU gates for computation. But this doubling is not visible anywhere! Page 23 to 24, the argument gets even more difficult to follow : I do not understand the point that is being made at the top of page 23 that there is somehow a difference between stating what the 2 hidden neurons are and what they compute. The induction is only more hairy and there is hardly any clear proof anywhere about why this composed function should be seen as computing the required function! Maybe the argument of Theorem 3 is correct and it does seem plausible but as it stands I am unable to decide the correctness given how unintelligible the presentation is! ============================================================== Response to "author response" : I thank the authors for their detailed response. I do think its a very interesting piece of work and after the many discussions with the co-reviewers now I am more convinced of the correctness of the proofs. Though still there are major sections in the paper like the argument about Lemma 9/Theorem 3 which are still too cryptic for me to judge correctness with full certainty. This paper definitely needs a thorough rewriting with better notation and presentation.

Reviewer 2



This paper presents a general framework for norm-based capacity control for L_{p,q} weight normalized deep fully connected networks with ReLu activation functions while considering the bias term in all layers, by establishing the upper bound on Radamacher complexity. For case p \geq 1, they need to use Radamacher average which results in dependency of upper bound on average width of the network and this is inevitable. Then with an L_{1,q} normalization they discuss the Next, the authors analyze the regression setting and provide approximation error as well as generalization error bound. In this case they argue that if you do L_{1,\infty} normalization of the weights, both generalization and approximation error will be bounded by L_1 norm of the output layer. In this case the upper bound does not depend on width and depth of the network, whereas if p >1, the upper bound will be a function of average width of the network. The authors analyze the binary classification and just mention the case for multi-class and how the bounds can change in one line (198). I would like to see more explanation. I find the analysis interesting and the proofs accurate and well-written. A downside of the paper is that the authors did not propose a practical algorithm to impose the weight normalization, specifically for L_{1,\infty} normalization. It is of interest to see how big c_0 is in practice. Can the authors argue about it theoretically? The next important thing then would be to show that the method works in practice (on a couple of datasets such as CIFAR100). However, I think the paper is worth getting accepted as it is provides the first step towards better regularization for deep networks. Minor point: In some places in the paper (line 13,18, 42,129), the fully connected DNNs are called feed forward DNNs. Note that, while in earlier days this was ok, it is better not to use it anymore since convnets are essentially feeding the input forward and this will cause confusion. Please use “fully connected” throughout the paper.

Reviewer 3



This paper examines the Rademacher complexity of L_{p,q} normalized ReLU neural networks. Results are given that imply that generalization and approximation error can be controlled by normalization. These results are interesting and represent a meaningful contribution to the understanding of ReLU networks. Some improvements do need to be made to the writing. In particular, the results are simply listed in the main text with all accompanying motivation or intuition deferred to the proofs in the supplement. Since the paper appears to have too much white space between lines, there should be plenty of room to provide reasonably detailed intuition and/or proof sketches for the main results in the main text. This addition would greatly improve the readability of the paper for a larger audience. Edit: The rebuttal addressed all issues I had, and the new proof they indicate will be in the final version is interesting. I therefore vote for accept.